
# Exact relaxation to Gibbs and non-equilibrium steady states in the quantum cellular automaton Rule 54

**Katja Klobas**[*], **Bruno Bertini**

Rudolf Peierls Centre for Theoretical Physics, Oxford University,
Parks Road, Oxford OX1 3PU, United Kingdom

[*] katja.klobas@physics.ox.ac.uk

## Abstract

We study the out-of-equilibrium dynamics of the quantum cellular automaton Rule 54 using a time-channel approach. We exhibit a family of (non-equilibrium) product states for which we are able to describe exactly the full relaxation dynamics. We use this to prove that finite subsystems relax to a one-parameter family of Gibbs states. We also consider inhomogeneous quenches. Specifically, we show that when the two halves of the system are prepared in two *different* solvable states, finite subsystems at finite distance from the centre eventually relax to the non-equilibrium steady state (NESS) predicted by generalised hydrodynamics. To the best of our knowledge, this is the first exact description of the relaxation to a NESS in an interacting system and, therefore, the first independent confirmation of generalised hydrodynamics for an inhomogeneous quench.

# 1 Introduction

Over the last two decades intense efforts by both experimentalists and theorists have greatly advanced our understanding of isolated quantum matter out of equilibrium [1–9]. It is now established that at asymptotically large times expectation values of local observables relax to time-independent numbers in translationally invariant systems [6,9], while they follow slow dynamics governed by emergent classical hydrodynamic equations [10–12] when the translational invariance is weakly broken. This surprising onset of relaxation in isolated systems is induced by the effective bath created by the system on its own parts, and the final (quasi) stationary state of the system is determined by the conservation laws with local densities [13].

In spite of the aforementioned great progress many questions remain wide open. In particular, it is still unclear how and when the slow hydrodynamic regime is reached and what is the role played by the local conservation laws in the relaxation process. This question is related to the more general problem of describing the dynamics of out-of-equilibrium quantum matter for large but finite times. This is arguably one of the key challenges of modern theoretical physics and no general method to address it is currently known. In particular, while generic interacting many-body systems are clearly out of the scope of analytical methods, available computational approaches can reach late times only by introducing drastic approximations of the microscopic dynamics [14–16]. Exact methods are limited to small systems [4, 17] or to initial states with ultra-short-ranged correlations [18]. Even in one dimension, where advanced techniques based on matrix product states [19–21] are able to tackle large systems in an exact fashion, numerical studies are limited by the rapid growth of entanglement and can only describe short times. Furthermore, continuum models — which describe many cold-atom experiments — provide even more serious challenges for the numerics.

Surprisingly, the situation is no better in the case of interacting *integrable* systems, i.e. systems characterised by an extensive number of local conservation laws. Indeed, even though integrability gives direct access to thermodynamics [22], it is generically of little help when it comes to address the finite-time dynamics in the presence of interactions. This is essentially due to the fact that the eigenstates of integrable models, although in principle known, have a very complicated structure [22], which prevents most practical manipulations. In fact, in

recent years a substantial number of non-trivial results concerning quantum many-body dynamics have not come from integrable systems, but rather from the opposite limiting case of minimally structured (or maximally chaotic) systems, i.e. systems without local conservation laws [23–39]. In particular, the so called dual-unitary circuits [40] have been shown to offer an invaluable testing ground where many dynamical quantities [40–49], including the time evolution of local observables [43,47], can be determined exactly.

Understanding the dynamics of integrable systems remains, however, of crucial importance, for instance it could unveil the role played by conservation laws in the relaxation. An interesting route to achieve this goal has recently been pointed out in Refs. [50–56], which showed that in some *special* integrable models one can indeed access the full time evolution of local observables. In particular Ref. [50] adopted a tensor network formulation to compute the full (local) dynamics of a class of "solvable" initial states in the quantum cellular automaton Rule 54 [57], up to their eventual relaxation to the infinite temperature state. The main observation leading to exact results has been that the dynamics of the system are simple when observed from the time channel, i.e. when propagating in space rather than in time.

Here we provide a highly non-trivial extension of these results computing the exact local dynamics of a larger class of initial states that, at infinite times, relax to non-trivial Gibbs ensembles. A very interesting consequence of our results is that we can study exactly settings originating non-trivial transport of conservation laws at asymptotically large times, when the system is expected to follow the prediction of *generalized hydrodynamics* (GHD) [58,59]. This gives the unprecedented possibility of testing this expectation. In particular, here we derive *ab initio* the prediction of GHD for the non-equilibrium steady state attained by the system following a bipartitioning protocol, i.e. the sudden junction of two halves of the system prepared in different homogeneous states.

This is the first of two papers devoted to the study of the dynamics of Rule 54 from the aforementioned extended family of solvable initial states. While here we consider the dynamics of local observables, in Ref. [60] (from now on Paper II) we study the growth of entanglement.

The rest of the paper is laid out as follows. In Sec. 2 we introduce the general time-channel approach that we adopt to find our results. In Sec. 3 we specialise the treatment to the case of Rule 54. In Sec. 4 we determine the extended family of solvable initial states and in Sec. 5 we present an exact solution of the quench dynamics. In Sec. 6 we compare our exact results with the asymptotic description of GHD and, finally, in Sec. 7 we report our conclusions. Several technical points and proofs are relegated to the two appendices.

## 2 Time-channel formulation of the local dynamics

In this section we introduce a general time-channel (or dual) description of the dynamics of local operators. It is based on the simple observation that the time evolution generated by a unitary matrix product operator (MPO) can be thought of as an evolution in space rather than in time. Even though this applies very widely — the evolution generated by a local Hamiltonian can be approximated arbitrary well by a unitary MPO [61,62] — it generically gives no clear computational advantage [42,63–65]. Interestingly, however, in certain cases this alternative picture becomes extremely powerful leading to exact results [40,41,43,44,47,50,66–70] and efficient computations [47,71–79].

Let us consider a one-dimensional chain of $2L$ qudits (with $d$ internal states) defined in the Hilbert space

$$\mathcal{H}_L = \bigotimes_{x=1}^{2L} \mathbb{C}^d \, , \tag{1}$$

and study the situation in which the system is driven out of equilibrium through a standard

quantum quench protocol [80, 81]. First it is prepared in a non-stationary state $|\Psi_0\rangle$ and then let to evolve according to a unitary propagator $\mathbb{U}$. Here we consider the case in which the initial state is a two-site shift invariant matrix product state (MPS)

$$|\Psi_0\rangle = \sum_{r_j \in \mathbb{Z}_d} \text{tr}\left[R^{r_1} S^{r_2} R^{r_3} \cdots S^{r_{2L}}\right] |r_1 r_2 \ldots r_{2L}\rangle \,, \tag{2}$$

where $R^r$ and $S^r$ are $\chi' \times \chi'$ matrices — $\chi'$ is commonly referred to as *bond dimension* — and the propagator has a staggered structure in time

$$\mathbb{U} = \mathbb{U}_o \mathbb{U}_e \,. \tag{3}$$

Moreover, we assume that the propagators for even and odd times are expressed in the following MPO-form

$$\mathbb{U}_e = \sum_{s_j, r_j \in \mathbb{Z}_d} \text{tr}\left[E^{s_1 r_1} F^{s_2 r_2} E^{s_3 r_3} \cdots F^{s_{2L} r_{2L}}\right] |r_1 r_2 \ldots r_{2L}\rangle\langle s_1 s_2 \ldots s_{2L}| \,,$$

$$\mathbb{U}_o = \sum_{s_j, r_j \in \mathbb{Z}_d} \text{tr}\left[F^{s_1 r_1} E^{s_2 r_2} F^{s_3 r_3} \cdots E^{s_{2L} r_{2L}}\right] |r_1 r_2 \ldots r_{2L}\rangle\langle s_1 s_2 \ldots s_{2L}| \,, \tag{4}$$

where $E^{sr}$ and $F^{sr}$, are $\chi \times \chi$ matrices (the bond dimension $\chi$ of those matrices is generically different from $\chi'$). The objects introduced above admit an intuitive graphical representation given respectively by

$$|\Psi_0\rangle = \underbrace{\;\;\triangleleft\!-\!\triangleright\!-\!\triangleleft\!-\!\triangleright\!-\!\triangleleft\!-\!\triangleright\!-\!\triangleleft\!-\!\triangleright\!-\!\triangleleft\!-\!\triangleright\!-\!\triangleleft\!-\!\triangleright\;\;}_{2L} \,, \tag{5}$$

and

$$\mathbb{U}_e = \underbrace{\;\cdots\!\circ\!-\!\bullet\!-\!\circ\!-\!\bullet\!-\!\circ\!-\!\bullet\!-\!\circ\!-\!\bullet\!-\!\circ\!-\!\bullet\!-\!\circ\!\cdots\;}_{2L} \,, \tag{6a}$$

$$\mathbb{U}_o = \underbrace{\;\cdots\!\bullet\!-\!\circ\!-\!\bullet\!-\!\circ\!-\!\bullet\!-\!\circ\!-\!\bullet\!-\!\circ\!-\!\bullet\!\cdots\;}_{2L} \,, \tag{6b}$$

where we introduced the tensors

$$\overset{s}{\underset{\triangleleft}{\big|}} = R^s \,, \qquad \overset{s}{\underset{\triangleright}{\big|}} = S^s \,, \qquad \underset{s}{\overset{r}{-\!\!\!\circ\!\!\!-}} = E^{sr} \,, \qquad \underset{s}{\overset{r}{-\!\!\!\bullet\!\!\!-}} = F^{sr} \,. \tag{7}$$

Note that the space-time staggering considered here is inessential and can be easily removed by fusing the two time-steps $\mathbb{U}_o$ and $\mathbb{U}_e$, and, at the same time, merging together two local sites. This results in a homogeneous MPO with larger bond dimension and qudits with more internal states. Here, however, we prefer to keep the staggering because it arises naturally in the quantum cellular automaton Rule 54 (see Sec. 3), which is the concrete example considered in this paper. We also remark that in the case of an MPO (3) describing *local* interactions (which is e.g. the case of Rule 54 – see Sec. 3), the tensors fulfil additional constraints. Since the upcoming discussion is largely independent of these constraints, we ignore them for the sake of simplicity. The only assumption that we make on the tensors (7) is

**Assumption 1.** *The state transfer matrix*

$$\tau = \sum_{s,r}(S^{s*}\otimes S^s)(R^{r*}\otimes R^r),\tag{8}$$

*has a unique maximal eigenvalue, which, without loss of generality, can be taken equal to one. Namely, the geometric and algebraic multiplicity of the eigenvalue one are equal to one.*

We remark that, since

$$\langle\Psi_0|\Psi_0\rangle = \mathrm{tr}\!\left[\tau^L\right],\tag{9}$$

the above assumption ensures that $|\Psi_0\rangle$ is normalised to one in the thermodynamic limit.

Let us consider the expectation value of a local operator $\mathcal{O}_x$ on the state at time $t$, where the subscript $x$ denotes the position of the left edge of the operator's support. Using our graphical representation, we can depict it as follows

$$\frac{\langle\Psi_t|\mathcal{O}_x|\Psi_t\rangle}{\langle\Psi_0|\Psi_0\rangle} = \frac{1}{\langle\Psi_0|\Psi_0\rangle}$$ 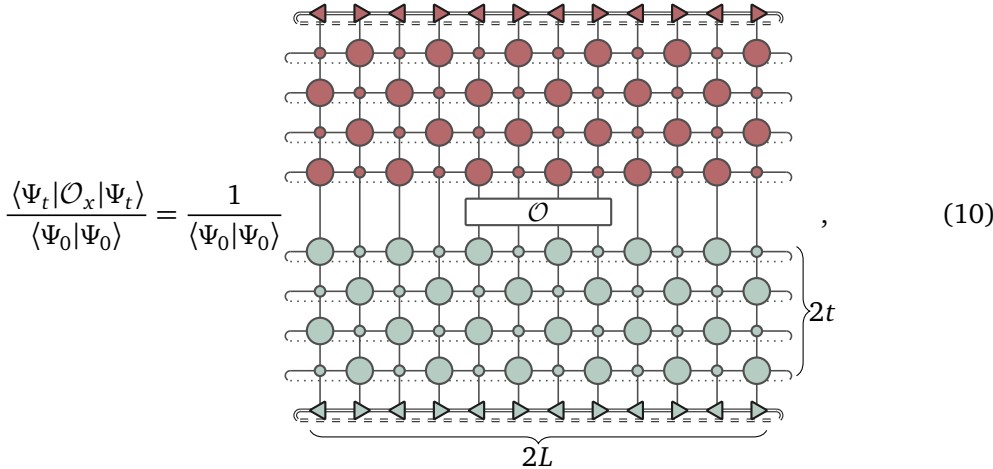 $$,\tag{10}$$

where we defined

$$|\Psi_t\rangle = \mathbb{U}^t\,|\Psi_0\rangle\,,\tag{11}$$

and introduced the symbols

$$\mbox{} = \mbox{}^{*},\qquad \mbox{} = \mbox{}^{*},\qquad \mbox{} = \mbox{}^{*},\qquad \mbox{} = \mbox{}^{*},\tag{12}$$

for the complex conjugate of the tensors (7).

We now interpret the tensor network (10) in terms of an evolution in the space direction [63–65]. Specifically, by defining *space transfer matrices* as

$$\tilde{\mathbb{W}} = \quad,\qquad \tilde{\mathbb{W}}[\mathcal{O}] = \quad,\tag{13}$$

we can rewrite Eq. (10) as

$$\frac{\langle \Psi_t | \mathcal{O}_x | \Psi_t \rangle}{\langle \Psi_0 | \Psi_0 \rangle} = \frac{\text{tr}\left( \tilde{\mathbb{W}}^{L - s_{\mathcal{O}}/2} \tilde{\mathbb{W}}[\mathcal{O}] \right)}{\langle \Psi_0 | \Psi_0 \rangle}, \tag{14}$$

where $s_{\mathcal{O}}$ is support of $\mathcal{O}$ (e.g. $s_{\mathcal{O}} = 4$ in (10) and (13)), which we conveniently take to be even.

To characterise the behaviour of the r.h.s. of (14) in the limit of large system sizes, we should understand the spectral properties of the space transfer matrix $\tilde{\mathbb{W}}$. The latter are summarised by the following general property.

**Property 1.** *Whenever the time evolution is unitary and the state transfer matrix has unique maximal eigenvalue* 1, *the transfer matrix* $\tilde{\mathbb{W}}$ (13) *has also a unique eigenvalue* $\lambda_0 = 1$ *while all other eigenvalues of* $\tilde{\mathbb{W}}$ *are strictly contained in the unit circle.*

*Proof.* The unitarity of time-evolution implies

$$1 = \frac{\langle \Psi_t | \Psi_t \rangle}{\langle \Psi_0 | \Psi_0 \rangle} = \frac{1}{\langle \Psi_0 | \Psi_0 \rangle} \text{tr}\left[ \tilde{\mathbb{W}}^L \right] = \frac{1}{\langle \Psi_0 | \Psi_0 \rangle} \sum_{j \geq 0} \lambda_j^L, \tag{15}$$

where the second equality follows from unitarity and the third one from the definition (13). Since

$$\lim_{L \to \infty} \langle \Psi_0 | \Psi_0 \rangle = 1 \tag{16}$$

in the limit $L \to \infty$ the above equality is satisfied only if $\lambda_0 = 1$ and $\left| \lambda_{j \geq 1} \right| < 1$. $\qquad\square$

As a consequence of Property 1 we have

$$\lim_{L \to \infty} \frac{\langle \Psi_t | \mathcal{O}_x | \Psi_t \rangle}{\langle \Psi_0 | \Psi_0 \rangle} = \frac{\langle L | \tilde{\mathbb{W}}[\mathcal{O}_x] | R \rangle}{\langle L | R \rangle}, \tag{17}$$

where we respectively denoted by $\langle L |$ and $| R \rangle$ the left and right leading eigenvectors of $\tilde{\mathbb{W}}$ (also referred to as *fixed points*), i.e. the vectors fulfilling

$$\langle L | \tilde{\mathbb{W}} = \langle L | , \qquad \tilde{\mathbb{W}} | R \rangle = | R \rangle . \tag{18}$$

Property 1 ensures that these vectors are unique up to a multiplicative constant. Since (17) holds for any local observable $\mathcal{O}$, we can represent the density matrix reduced to a subsystem $\mathcal{S}$ by means of the following diagram

$$\rho_{\mathcal{S}}(t) = \lim_{L \to \infty} \text{tr}_{\bar{\mathcal{S}}} | \Psi_t \rangle \langle \Psi_t | = \frac{1}{\langle L | R \rangle} \langle L | \quad | R \rangle , \tag{19}$$

where blue rectangles denote $\langle L |$ and $| R \rangle$ and $\bar{\mathcal{S}}$ indicates the complement of $\mathcal{S}$.

The above equations give us an intuitive interpretation of fixed-points: they describe quantitatively the effective bath created by $\bar{\mathcal{S}}$ in the thermodynamic limit (with $\mathcal{S}$ remaining finite). The emergence of such an effective bath is what allows $\mathcal{S}$ to reach a stationary state, see e.g. Refs. [6, 42, 50].

The practical utility of the representations (17) and (19) in determining the relaxation dynamics of $\mathcal{S}$ depends on the form of the fixed points. For instance, they become extremely useful when the fixed points can be represented as MPSs with a constant (in time) bond dimension. Indeed, as demonstrated in Ref. [50], in this case the full dynamics of any local observable can be accessed by diagonalising a finite-dimensional matrix.

The bond dimension of the fixed points directly reflects the nature of the effective bath. Specifically, when the bath is Markovian the fixed points become product states [42]. This is expected to occur at large times in systems with no local conservation laws. For intermediate times, however, the bond dimension of the fixed points is typically observed to grow exponentially [63, 64]. An important exception are dual-unitary circuits evolving from solvable states, where the bath is Markovian for all times [43, 67].

The situation is even more complicated in integrable systems: since the bath is never expected to become Markovian, the fixed points have no reason to be simple even for large times. Nevertheless, approaches based on the space transfer matrix (also called quantum transfer matrix in this context) have a relative long history in the literature of integrable models [82–84]. In particular, Refs. [68–70] have proposed a programme aiming at combining time channel approaches with Bethe Ansatz to access the finite time dynamics. Up to now, however, this only led to the calculation of the so-called Loschmidt echo, which is easier to treat but less physically transparent than, for instance, local observables or entanglement.

Remarkably, as we discuss below, Rule 54 represents an exception. Even though the system is integrable, there exist a class of initial states for which its fixed points are simple (MPSs with bond dimension three) for all times [50].

Before concluding this general discussion we will show that, besides homogeneous quantum quenches, the formalism described here can be directly applied to two additional physically relevant quench problems

- *Bipartitioning protocols* [10, 11, 85, 86], i.e. quenches from inhomogeneous states that are composed by joining two different homogeneous pieces (or "leads").

- Dynamical correlation functions in a stationary state, i.e. *local quenches*.

## 2.1 Bipartitioning protocols

In the case of bipartitioning protocols one considers an initial state of the form

$$|\Psi_0\rangle = \qquad\qquad\qquad , \tag{20}$$

where

$$\tag{21}$$

and

$$\tag{22}$$

are different tensors (all fulfilling Assumption 1). Repeating the steps above we have that the expectation value of an observable at distance $x \geq 0$ from the junction is given by

$$\frac{\langle \Psi_t | \mathcal{O}_x | \Psi_t \rangle}{\langle \Psi_0 | \Psi_0 \rangle} = \frac{\text{tr}\left( \tilde{\mathbb{W}}_{\text{L}}^{L/4} \tilde{\mathbb{W}}_{\text{R}}^{x/2} \tilde{\mathbb{W}}_{\text{R}}[\mathcal{O}] \tilde{\mathbb{W}}_{\text{R}}^{(L-2s_{\mathcal{O}}-2x)/4} \right)}{\langle \Psi_0 | \Psi_0 \rangle} , \tag{23}$$

where we introduced the space transfer matrices of the two leads

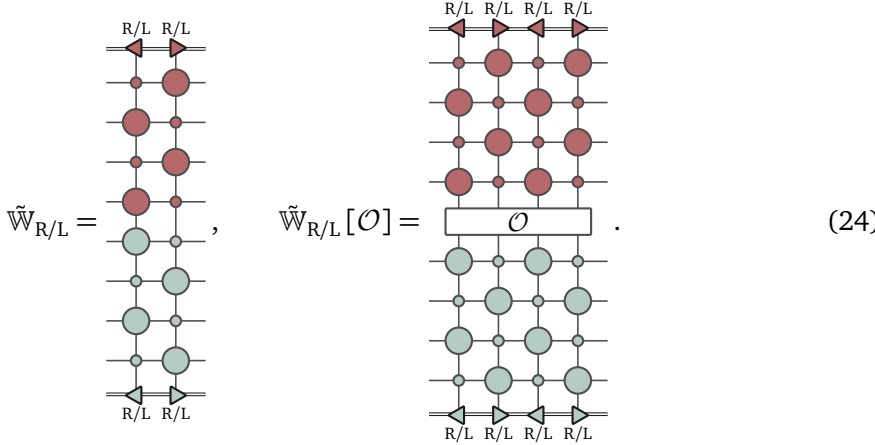

$$\tilde{\mathbb{W}}_{\text{R/L}} = \quad , \quad \tilde{\mathbb{W}}_{\text{R/L}}[\mathcal{O}] = \quad . \tag{24}$$

Considering the thermodynamic limit, for finite $x \geq 0$ we find

$$\lim_{L \to \infty} \frac{\langle \Psi_t | \mathcal{O}_x | \Psi_t \rangle}{\langle \Psi_0 | \Psi_0 \rangle} = \frac{\langle L_{\text{L}} | \tilde{\mathbb{W}}_{\text{R}}^{x/2} \tilde{\mathbb{W}}_{\text{R}}[\mathcal{O}] | R_{\text{R}} \rangle}{\langle L_{\text{L}} | R_{\text{R}} \rangle} , \tag{25}$$

where the normalisation can be fixed by choosing $\mathcal{O}_x$ equal to the identity operator. This gives access to the reduced density matrix of any finite subsystem at distance $x$ from the junction

$$\rho_{\mathcal{S},x}(t) = \frac{1}{\langle L_{\text{L}} | R_{\text{R}} \rangle} \langle L_{\text{L}} | \quad | R_{\text{R}} \rangle . \tag{26}$$

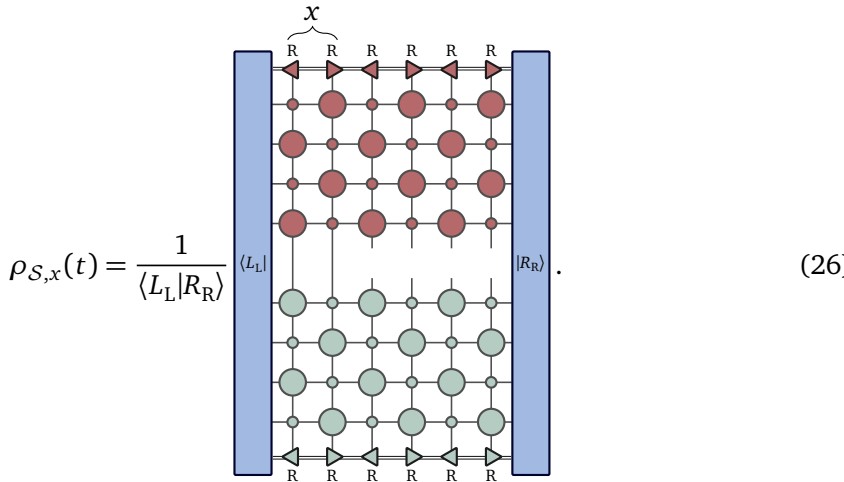

## 2.2 Dynamical correlations at equilibrium

The dynamical correlation function between two generic local observables $\mathcal{O}_{1,0}$ and $\mathcal{O}_{2,x}$ in a stationary state $\rho_{\mathrm{s}}$ can be represented by means of the following diagram

$$\mathrm{tr}\left(\rho_{\mathrm{s}}\mathcal{O}_{1,0}\mathbb{U}^{-t}\mathcal{O}_{2,x}\mathbb{U}^{t}\right) = \frac{1}{Z} \quad \text{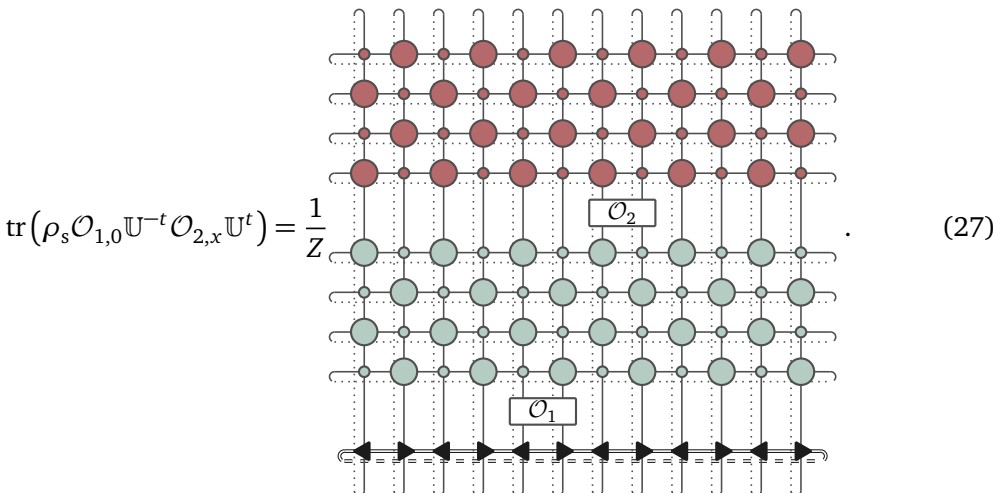} \quad . \tag{27}$$

Here we considered a stationary state written as an MPO with the same (two-site) translational symmetry as the time-evolution operator, i.e. we represented it as

$$\rho_{\mathrm{s}} = \frac{1}{Z} \quad , \tag{28}$$

where $Z$ is the normalisation. We again assume that the state transfer matrix of $\rho_{\mathrm{s}}$, i.e.

$$\tau_{\mathrm{s}} = \quad , \tag{29}$$

has unique maximal eigenvalue one. Therefore

$$\lim_{L\to\infty} Z = \lim_{L\to\infty} \mathrm{tr}\left[\tau^{L}\right] = 1 . \tag{30}$$

As before, we introduce the transfer-matrix in the space direction

$$\tilde{\mathbb{W}}_{\mathrm{s}} = \quad , \tag{31}$$

which fulfils Property 1 (the proof is completely analogous). This implies that, in the thermodynamic limit, the correlation function can again be expressed as a finite tensor-network with

the fixed-points $\langle L_{\mathrm{s}}|$, $|R_{\mathrm{s}}\rangle$ on the left and right edge

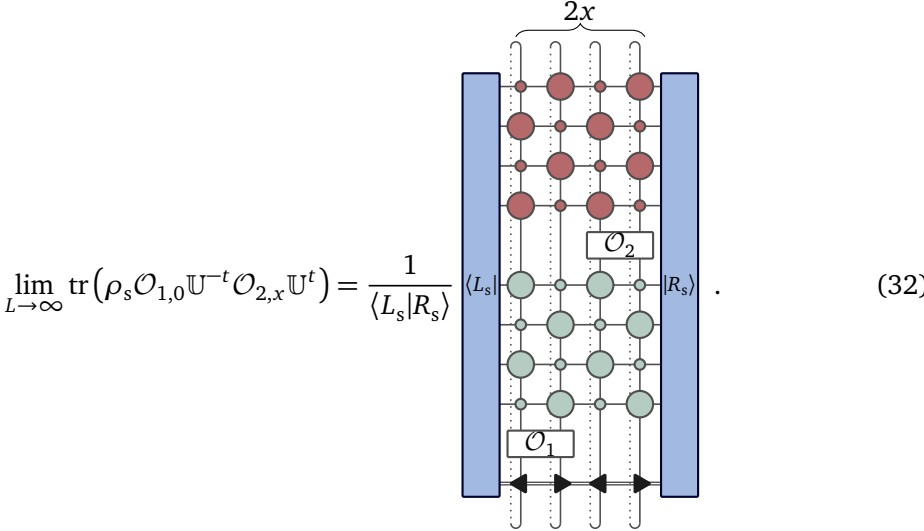

$$\lim_{L\to\infty} \mathrm{tr}\left(\rho_{\mathrm{s}}\mathcal{O}_{1,0}\mathbb{U}^{-t}\mathcal{O}_{2,x}\mathbb{U}^{t}\right) = \frac{1}{\langle L_{\mathrm{s}}|R_{\mathrm{s}}\rangle} \quad . \tag{32}$$

## 3 A solvable case: quantum cellular automaton Rule 54

In this paper we adopt the time channel approach summarised in the previous section to describe the non-equilibrium dynamics of a specific integrable system: the reversible cellular automaton given by the Rule 54 in the classification of Ref. [87] (it corresponds 250R in the earlier classification of Ref. [88]), which can be seen as a deterministic discrete-time limit of the Fredrickson-Andersen model [89]. In recent years, this model has been recognised as one of the simplest examples of interacting integrable systems, where many non-equilibrium properties can be described exactly, both in the classical [90–97] (see also a recent review [57]), and in the quantum realm [50, 98–102]. The integrability of the model was conjectured already in [87], and later confirmed in Ref. [100], which derived its Bethe Ansatz equations. However, many exact results obtained in this model (including the ones discussed here) go beyond what is possible for typical interacting integrable systems and, furthermore, they do not explicitly use any integrability-related property.

The model can be defined as a local — brickwork-like — quantum circuit on qubits. In this system, however, the gates are not applied in the standard two-site shift invariant pattern but act non-trivially on three consecutive sites, see Fig. 1. More specifically, the system is defined in a periodic chain of $2L$ qubits with Hilbert space

$$\mathcal{H}_L = \bigotimes_{x=1}^{2L} \mathbb{C}^2 \,, \tag{33}$$

and $\{|s_x\rangle_x\}_{s_x=0,1}$ denotes the standard computational basis (i.e. the basis of eigenstates of all $\{\sigma_{3,x}\}_{x\in\mathbb{Z}_{2L}}$, where $\sigma_{3,x}$ is the third Pauli matrix at site $x$). The time evolution is discrete and generated by the unitary operator

$$\mathbb{U} = \mathbb{U}_{\mathrm{e}}\mathbb{U}_{\mathrm{o}} \,, \tag{34}$$

with

$$\mathbb{U}_{\mathrm{o}} = \prod_{x\in\mathbb{Z}_L} U_{2x+1} \,, \qquad \mathbb{U}_{\mathrm{e}} = \prod_{x\in\mathbb{Z}_L} U_{2x} \,. \tag{35}$$

Here we introduced the notation

$$U_x = \mathbb{1}^{\otimes(x-1)} \otimes U \otimes \mathbb{1}^{\otimes(2L-x-2)} \,, \tag{36}$$

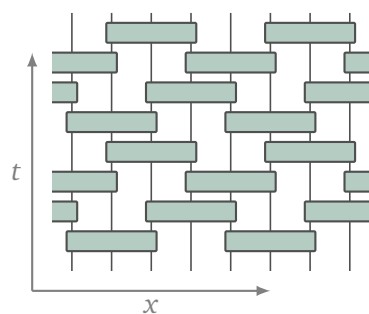

Figure 1: Graphical representation of the time evolution operator up to time $t = 2$.

where $\mathbb{1}$ denotes the identity operator on one qubit and $U$ is the three-site local gate defined by the following matrix elements in the computational basis

$$U^{s'_1 s'_2 s'_3}_{s_1 s_2 s_3} = \quad = \delta_{s_1,s'_1} \delta_{\chi(s_1,s_2,s_3),s'_2} \delta_{s_3,s'_3}, \tag{37}$$

where the "updated value" of the middle site is determined by

$$\chi(s_1, s_2, s_3) \equiv s_1 + s_2 + s_3 + s_1 s_3 \quad (\text{mod } 2). \tag{38}$$

Note that, since $U_x$ and $U_{x+2}$ commute, the ordering of the products (35) is inessential.

The time evolution operators for even and odd times (cf. (35)) can be expressed in the MPO form (4), by identifying tensors (7) as

$$s_1 \; \rule{0pt}{0pt} \bigcirc \; s_3 = \delta_{\chi(s_1,s_2,s_3),s_4}, \qquad s_1 \; \rule{0pt}{0pt} \; s_3 = \prod_{j=1}^{3} \delta_{s_j,s_{j+1}}. \tag{39}$$

To establish the equivalence between (4) and (34,35) we generalise the definition of the "small circle" tensor to $k$ legs

$$s_1 \; \rule{0pt}{0pt} \;\; = \prod_{j=1}^{k-1} \delta_{s_j,s_{j+1}}. \tag{40}$$

Now we are able to express $U$ in terms of small and big circles as

$$U = \quad = \quad, \tag{41}$$

and the equivalence follows immediately using

$$= \quad = \quad. \tag{42}$$

# 4 Left and right fixed points in Rule 54

Using the fact that the MPO (3) is *local* for Rule 54 (i.e. it can be equivalently represented in terms of mutually commuting local unitary gates (35)), we can immediately express the left and right fixed points of the space transfer-matrix $\tilde{\mathbb{W}}$ (13) for *any* initial MPS fulfilling Assumption 1 as

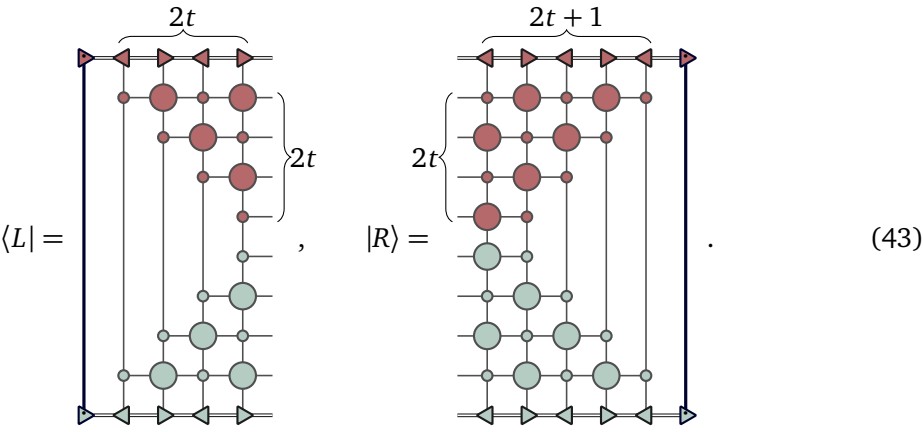

$$\langle L| = \qquad , \qquad |R\rangle = \qquad . \tag{43}$$

where

$$\qquad , \qquad , \tag{44}$$

are respectively the left fixed point of $\tau$ (cf. (8)), and the right fixed point of

$$\tau' = \qquad , \tag{45}$$

which differs from $\tau$ because of the two triangular tensors being swapped. Analogously, the fixed points of $\tilde{\mathbb{W}}_s$ are expressed as

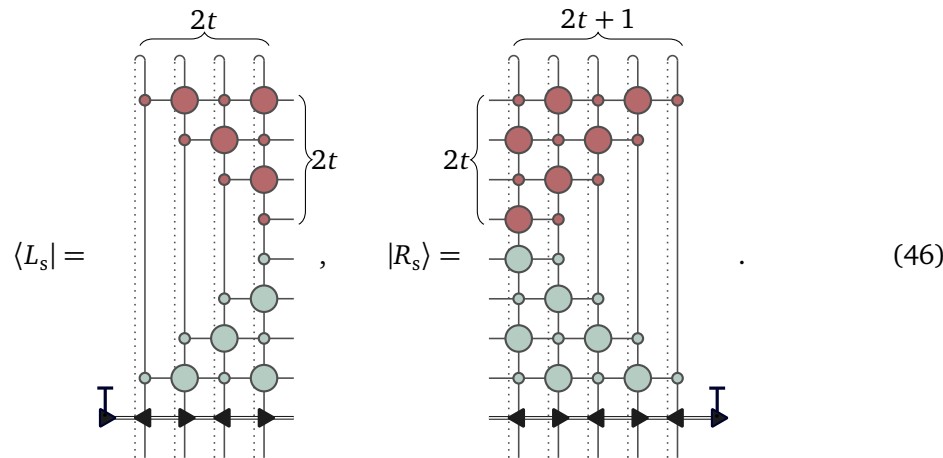

$$\langle L_s| = \qquad , \qquad |R_s\rangle = \qquad . \tag{46}$$

where we took $\tau_s$ (cf. (29)) and

$$\tau'_s = \qquad , \tag{47}$$

with a unique maximal eigenvalue 1 and left and right fixed points given by

$$\qquad , \qquad . \tag{48}$$

We can immediately verify that $\langle L|$ in (43) is indeed a left eigenvector of $\tilde{\mathbb{W}}$ corresponding to eigenvalue 1

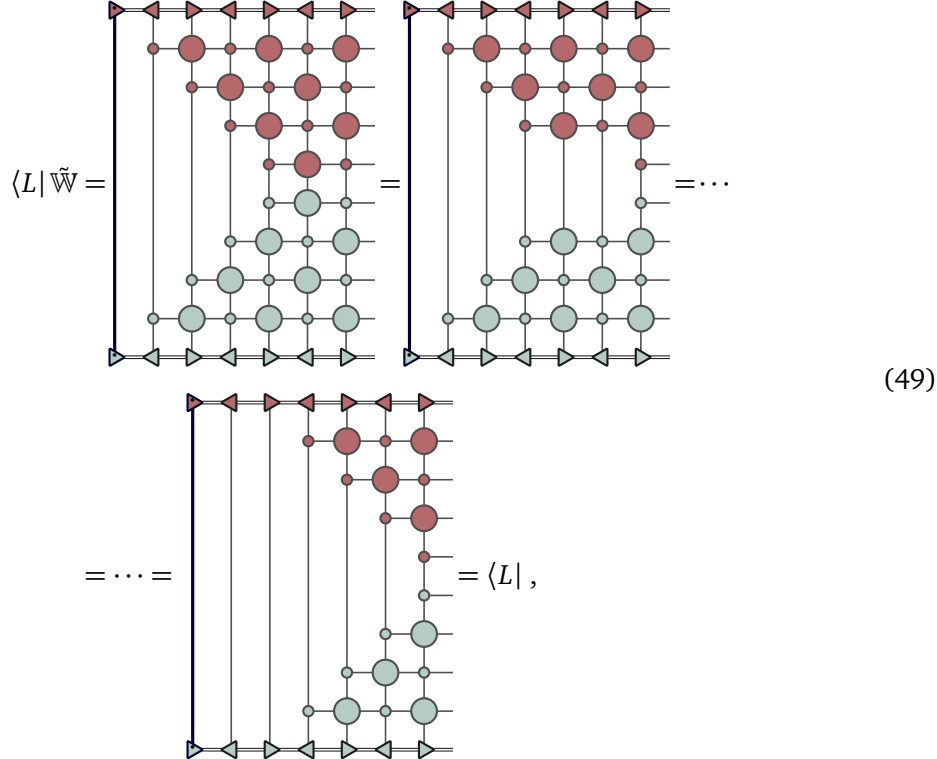

$$(49)$$

where we repeatedly use the unitarity of $U$, i.e.

$$(50)$$

together with the identity (42). In the last step we also used that the first of (44) is the left fixed point of $\tau$. The proof of the relations

$$\tilde{\mathbb{W}}|R\rangle = |R\rangle\,, \qquad \langle L_s|\tilde{\mathbb{W}}_s = \langle L_s|\,, \quad \text{and} \quad \tilde{\mathbb{W}}_s|R_s\rangle = |R_s\rangle\,, \qquad (51)$$

is completely analogous.

The general form of the fixed points suggests a more convenient diagrammatic representation obtained by bending the top half of tensor networks behind the bottom half and introducing *folded* tensors [63]

$$(52)$$

where the local Hilbert space on which these objects act is effectively *doubled* — it corresponds to two qubits rather than 1. Using the folded representation, the space transfer matrices $\tilde{\mathbb{W}}$ and $\tilde{\mathbb{W}}_s$, defined in Eqs. (13) and (31) respectively, take the form

$$\tilde{\mathbb{W}} = \left. \begin{matrix} \end{matrix} \right\}2t\,, \qquad \tilde{\mathbb{W}}_s = \left. \begin{matrix} \end{matrix} \right\}2t\,, \qquad (53)$$

while their fixed-points can be written as

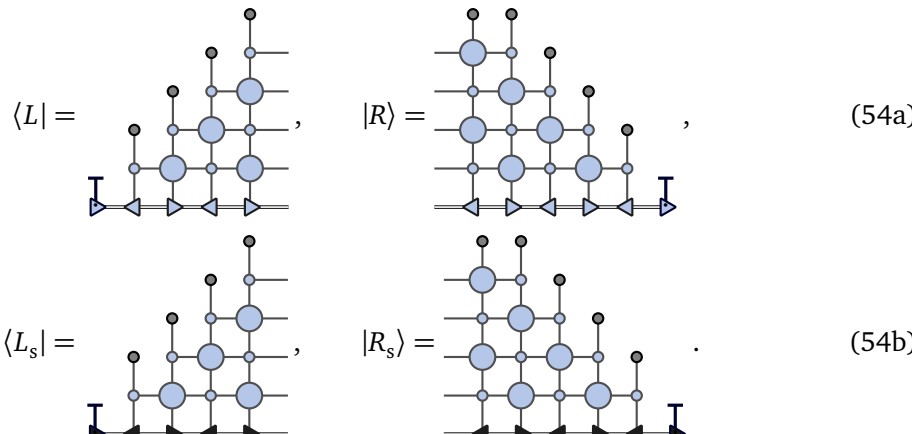

$$\langle L| = \qquad , \qquad |R\rangle = \qquad , \qquad (54a)$$

$$\langle L_{\mathrm{s}}| = \qquad , \qquad |R_{\mathrm{s}}\rangle = \qquad . \qquad (54b)$$

 The fixed-points given by the diagrams (54) generically exhibit bond dimension that grows exponentially with time $t$. However, as we argue below (and part of it was shown in [50]), in Rule 54 we can identify initial states and stationary states for which (54) reduce to an MPS with *constant* bond dimension $\chi = 3$.

## 4.1 Efficient MPS-representation of the fixed points

Here we identify a class of initial states (5) and stationary states (28) for which the tensor networks (54) can be simplified. We begin by writing the following MPS ansatz

$$\langle L_{\mathrm{A}}| = \qquad , \qquad |R_{\mathrm{A}}\rangle = \qquad , \qquad (55)$$

where $\langle L_{\mathrm{A}}|$ and $|R_{\mathrm{A}}\rangle$ can be either fixed points of $\tilde{\mathbb{W}}$ or of $\tilde{\mathbb{W}}_{\mathrm{s}}$ and we introduced the (so far unknown) "bulk tensors"

$$\qquad , \qquad , \qquad , \qquad (56a)$$

$$\qquad , \qquad , \qquad , \qquad (56b)$$

and "boundary vectors"

$$\top , \qquad (57a)$$

$$\qquad , \qquad , \qquad (57b)$$

$$\qquad , \qquad . \qquad (57c)$$

Next, we prove the ansatz (55) in two steps.

  (i) Find a set of *local* algebraic conditions for the tensors (56) and (57) that ensure invariance of (55) under the left/right action of space transfer matrix.

(ii) Solve them to find explicit representations of the tensors.

The step (ii) can be achieved only for certain initial or stationary states: these will form the "solvable" class.

To identify the solvable class it is useful to first consider the case of stationary states. Then, for a given solvable stationary state, we will find a corresponding family of solvable initial states with fixed points described by the same tensors (56) (but different boundary vectors (57)). Intuitively this means that we will search for a family of initial states that relaxes to a given solvable stationary state.

## 4.2 Solvable stationary states

### 4.2.1 Infinite temperature state

It is instructive to begin by reviewing the construction presented in Ref. [50] for the fixed points corresponding to the infinite temperature state. Namely we consider

$$\rho_\infty = \frac{1}{2^{2L}} \mathbb{1}^{\otimes 2L} = \frac{1}{2^{2L}} \overbrace{\downarrow \downarrow \downarrow \downarrow \downarrow \downarrow \downarrow \downarrow \downarrow \downarrow \downarrow \downarrow}^{2L}, \tag{58}$$

where the tensor

$$\downarrow = \uparrow^\dagger \tag{59}$$

is nothing but the identity operator in the folded representation. The space transfer matrix (53) corresponding to $\rho_\infty$ reads as

$$\mathbb{W}_\infty = \frac{1}{4} \left. \begin{matrix} \end{matrix} \right\} 2t \ . \tag{60}$$

Since the top and bottom boundary vectors for this matrix are the same we simplify the Ansatz (55) and consider

$$\langle L_A | = \quad , \qquad |R_A\rangle = \quad , \tag{61}$$

where

$$\bot = \mathsf{T}^\dagger. \tag{62}$$

At this point we note that if the tensors in (56) and (57) satisfy the following set of algebraic relations

$$\frac{1}{2} \quad = \quad , \qquad \frac{1}{2} \quad = \quad , \tag{63a}$$

$$\quad = \quad , \qquad \quad = \quad , \qquad \quad = \quad , \tag{63b}$$

we have

$$\langle L_{\rm A}|\,\tilde{\mathbb{W}}_\infty = \frac{1}{4}\,\,\,\,\,\,\, = \frac{1}{2}\,\,\,\,\,\,\, = \,\,\,\,\,\,\, = \langle L_{\rm A}|\,. \tag{64}$$

Namely $\langle L_{\rm A}|$ is the left fixed point of $\tilde{\mathbb{W}}_\infty$. To establish (64) we apply the first identity in (63a), which creates the 2-site tensor

$$\tag{65}$$

We then repeatedly move it upwards by the first of (63b) until it is absorbed at the top by the application of the second relation in (63b). The second step proceeds analogously starting with the second equality in (63a) and finishing with the last one of (63b).

The above construction shows that if one can find some tensors (56a) and (57a) solving (63) for some given small and large circles (cf. (52)), then the state $\langle L_\infty|$ (cf. (61)) is the left fixed point of $\tilde{\mathbb{W}}_\infty$. Remarkably, when big and small circles are the time-evolution tensors of Rule 54 (cf. (39)) the relations (63) admit a solution with bond dimension 3

$$\diamond \mapsto \blacklozenge\,, \qquad \square \mapsto \blacksquare\,, \qquad \square \mapsto \blacksquare\,, \qquad \mathsf{T} = \begin{bmatrix} 1 & 1 & 0 \end{bmatrix}. \tag{66}$$

In particular the one-site blue tensors are given by

$$\blacklozenge\,00 = \frac{1}{2}\begin{bmatrix} 1 & 1 & -1 \\ 1 & 1 & 1 \\ 1 & -1 & -1 \end{bmatrix}, \qquad \blacklozenge\,01 = \blacklozenge\,10 = \frac{1}{2}\begin{bmatrix} 0 & 1 & -1 \\ 1 & 0 & 0 \\ 1 & 0 & 0 \end{bmatrix},$$

$$\blacklozenge\,11 = \begin{bmatrix} 0 & 1 & 0 \\ 1 & 0 & 0 \\ 0 & 0 & 0 \end{bmatrix}, \qquad \blacksquare\,rs = \begin{bmatrix} \delta_{s,0}\delta_{r,0} & 0 & 0 \\ 0 & \delta_{s,1}\delta_{r,1} & 0 \\ 0 & 0 & \delta_{s,1}\delta_{r,1} \end{bmatrix}, \tag{67}$$

while the two-site one is reported in Eq. (148) of Appendix A.

Finally we note that, since (39) are symmetric under left-right flips, if the left tensors (56a) fulfil (63), then

$$\diamond \equiv \blacklozenge\,, \qquad \square \equiv \blacksquare\,, \qquad \square \equiv \blacksquare\,, \tag{68}$$

fulfil

$$\frac{1}{2}\,\,\,\,\,\,\, = \,\,\,\,\,\,\, , \qquad \frac{1}{2}\,\,\,\,\,\,\, = \,\,\,\,\,\,\, , \tag{69a}$$

$$\,\,\,\,\,\,\, = \,\,\,\,\,\,\, , \qquad \,\,\,\,\,\,\, = \,\,\,\,\,\,\, , \qquad \,\,\,\,\,\,\, = \,\,\,\,\,\,\, , \tag{69b}$$

This immediately implies that the state $|R_A\rangle$ (cf. (61)) built with the tensors (68) is a right fixed point of $\tilde{\mathbb{W}}_\infty$. Namely

$$\tilde{\mathbb{W}}_\infty |R_A\rangle = |R_A\rangle . \tag{70}$$

This gives an explicit expression of both fixed points corresponding to the infinite temperature state.

### 4.2.2 GGEs

The above construction can be generalised to fixed points of transfer matrices corresponding to the following family of *generalised Gibbs ensembles* (GGE)s

$$\rho_{\text{GGE}} = \frac{e^{-\mu_- N_- - \mu_+ N_+}}{\text{tr}(e^{-\mu_- N_- - \mu_+ N_+})}, \tag{71}$$

where

$$N_+ = \sum_{x\in\mathbb{Z}_L} P^-_{2x} P^-_{2x+1} + \sum_{x\in\mathbb{Z}_{2L}} P^+_x P^-_{x+1} P^+_{x+2},$$
$$N_- = \sum_{x\in\mathbb{Z}_L} P^-_{2x-1} P^-_{2x} + \sum_{x\in\mathbb{Z}_{2L}} P^+_x P^-_{x+1} P^+_{x+2}, \qquad \text{where} \quad P^\pm := \frac{\mathbb{1}\pm\sigma_3}{2}, \tag{72}$$

are the conserved charges corresponding respectively to the number of left and right moving solitons, while $\mu_\pm$ are the associated chemical potentials.

The state in (71) exhibits a staggered MPO representation (28) with bond dimension 3 (see e.g. [96,97]). In the folded representation the latter reads as

$$\rho_{\text{GGE}} = \frac{1}{Z} \, \text{} , \tag{73}$$

with

$$\overset{s,\,r}{\underset{\blacktriangleright}{\big\downarrow}} = \delta_{s,r} W_s(z_-, z_+), \qquad \overset{s,\,r}{\underset{\blacktriangleleft}{\big\downarrow}} = \delta_{s,r} W'_s(z_-, z_+), \tag{74}$$

where

$$z_\pm = e^{-\mu_\pm}, \tag{75}$$

are the fugacities of left and right movers, and the $3 \times 3$ matrices $W_s(z,w), W'_s(z,w)$ are given by

$$W_0(z,w) = \begin{bmatrix} 1 & 0 & 0 \\ z & 0 & 0 \\ 1 & 0 & 0 \end{bmatrix}, \qquad W_1(z,w) = \begin{bmatrix} 0 & z & 0 \\ 0 & 0 & 1 \\ 0 & 0 & w \end{bmatrix}, \quad W'_s(z,w) = \frac{W_s(w,z)}{\lambda(w,z)}. \tag{76}$$

Here we introduced $\lambda(z,w)$ such that the state transfer matrix (cf. (29)) has maximal eigenvalue equal to one. This means that

$$\lambda \equiv \lambda(z_-, z_+), \tag{77}$$

is given by the largest solution to the following cubic equation

$$x^3 - (1 + 3z_- z_+)x^2 + (3z_-^2 z_+^2 - z_- z_+ - z_- - z_+)x - z_- z_+ (z_- z_+ - 1)^2 = 0. \tag{78}$$

Using the MPS representation (73) we can now formulate the algebraic relations for the tensors constituting the fixed points of the space transfer matrix

$$\tilde{\mathbb{W}}_s = \text{} , \tag{79}$$

with bottom boundary vectors given in Eq. (74). For definiteness, let us begin considering the conditions for the left fixed point in (55). We note that, since the *bulk relations* (63b) do not depend on the state at the bottom, they can be imposed also in the current case while we replace the *boundary relations* (63a) by

$$ \text{(80)} $$

Here, once again, the grey boundary vectors are given in Eq. (74). Using the same reasoning as below Eq. (64) we find that, if the tensors (56) and (57) satisfy (63b) and (80), then

$$ \langle L_A | \, \widetilde{\mathbb{W}}_s = \quad\quad = \quad\quad = \quad = \langle L_A | \, . \tag{81} $$

Therefore, to find an explicit representation of the fixed point we just have to solve (63b) and (80).

This can be done by realising that the bulk relations Eq. (63b) exhibit a one-parameter family of solutions with bond dimension 3

$$ \diamond \mapsto \blacklozenge_{\vartheta}, \quad \square \mapsto \blacksquare_{\vartheta}, \quad \square \mapsto \blacksquare_{\vartheta}, \quad \mathsf{T} = \begin{bmatrix} 1 & 1 & 0 \end{bmatrix}, \quad \vartheta \in [0,1]. \tag{82} $$

In particular, we have

$$ \blacklozenge_{\vartheta}\text{-}00 = \begin{bmatrix} 1-\vartheta & 1-\vartheta & -(1-\vartheta) \\ \vartheta & \vartheta & 1-\vartheta \\ \vartheta & -\dfrac{\vartheta^2}{1-\vartheta} & -\vartheta \end{bmatrix}, \qquad \blacklozenge_{\vartheta}\text{-}10 = \blacklozenge_{\vartheta}\text{-}01 = \begin{bmatrix} 0 & 1-\vartheta & -(1-\vartheta) \\ \vartheta & 0 & 0 \\ \vartheta & 0 & 0 \end{bmatrix}, $$

$$ \blacklozenge_{\vartheta}\text{-}11 = \begin{bmatrix} 0 & 1 & 0 \\ 1 & 0 & 0 \\ 0 & 0 & 0 \end{bmatrix}, \qquad\qquad \blacksquare_{\vartheta}\text{-}rs = \begin{bmatrix} \delta_{r,0}\delta_{s,0} & 0 & 0 \\ 0 & \delta_{r,1}\delta_{s,1} & 0 \\ 0 & 0 & \delta_{r,1}\delta_{s,1} \end{bmatrix}, \tag{83} $$

while the corresponding two-site tensors are reported in Eq. (148) of Appendix A. The infinite-temperature solution (67) is recovered for $\vartheta = 1/2$. In the above diagrams we explicitly reported $\vartheta$ to signal the dependence on this parameter. In the following, however, whenever the choice of $\vartheta$ is unambiguous we will ease the notation by removing it.

Plugging now (83) into (80) we can then solve for the left boundary vectors (57b) and for $\vartheta$. This admits a unique solution

$$ \vartheta \mapsto \vartheta_+ \equiv \frac{z_+(\lambda(1+z_-)+z_-(1-z_+z_-))}{\lambda(z_++z_--z_-z_+)}, \qquad \triangleright\!\!=\; \mapsto \; \triangleright\!\!-, \qquad \triangleleft\!\!=\; \mapsto \; \triangleleft\!\!-, \tag{84} $$

where the left boundary vectors $\triangleleft\!\!-$, $\triangleright\!\!-$ are reported in Eq. (149) of Appendix A.

The local relations for the *right* fixed point are again obtained by flipping (63b) and (80) to the left. Namely we consider (69b) and

$$\text{(85)}$$

Eq. (69b) are again solved by (68) with the blue tensors given in Eq. (83) and Eq. (148) of Appendix A. Plugging now into (85) and solving for the right boundary vectors and $\vartheta$ we find a unique solution

$$\vartheta \mapsto \vartheta_- \equiv \frac{z_-(\lambda(1+z_+)+z_+(1-z_+z_-))}{\lambda(z_++z_--z_+z_-)}, \qquad \qquad \text{(86)}$$

where the explicit expression for right boundary vectors is reported in Eq. (150) of Appendix A. Finally, we note that the mapping between $z_\pm$ and $\vartheta_\pm$ can be inverted

$$z_- = \frac{\vartheta_-(1-\vartheta_+)}{(1-\vartheta_-)^2}, \qquad z_+ = \frac{\vartheta_+(1-\vartheta_-)}{(1-\vartheta_+)^2}, \qquad \lambda = \frac{1}{(1-\vartheta_-)(1-\vartheta_+)}, \qquad \text{(87)}$$

which implies that the GGE can be equivalently parametrised by a pair of independent parameters $\vartheta_\pm \in [0,1]$. The choice $\vartheta_- = \vartheta_+$ corresponds to the GGE without an imbalance of particles, i.e. $\mu_- = \mu_+$.

## 4.3  Solvable initial states

Let us now consider *solvable initial states*, i.e. initial states for which the fixed points of the space transfer matrix

$$\tilde{\mathbb{W}} = \qquad \qquad \text{(88)}$$

are of the form (55). As anticipated in Sec. 4.1, we require these (pure) states to relax to the class of solvable GGEs discussed in the previous section. To this aim, we impose again the conditions (63b) and (69b) on the tensors (56) and (57) of the MPS ansatz. However, we replace the boundary relations (63a) and (69a) with

$$\text{(89)}$$

and

$$\text{(90)}$$

respectively. We remark that the difference between (63a, 69a) and (89, 90) is that the boundary vectors are now

$$
\text{(diagrams)} \qquad \text{(diagrams)}, \tag{91}
$$

i.e. they are formed by the tensor product of the tensors of the initial MPS (cf. (5)).

Once again the bulk relations (63b) and (69b) are solved by the family (83). In general left and right tensors are parametrised by different $\vartheta$s, which we denote by $\vartheta_+$ and $\vartheta_-$. Solving now separately the left relations using bulk tensors of the form (83) we find a solution for MPS matrices (91) of bond dimension *one*, i.e. for initial states in product form. Explicitly we have

$$
\text{(diagram)} \mapsto \text{(diagram)} \equiv \begin{bmatrix} 1 \\ 0 \\ 0 \end{bmatrix}, \qquad\qquad \text{(diagram)} \mapsto \text{(diagram)} \equiv \begin{bmatrix} 1-\vartheta_+ \\ \vartheta_+ \\ -\vartheta_+^2(1-\vartheta_+)^{-1} \end{bmatrix}, \tag{92a}
$$

$$
\text{(diagram)} \mapsto \text{(diagram)} \equiv \begin{bmatrix} e^{i\phi_1} \\ 0 \end{bmatrix}, \qquad\qquad \text{(diagram)} \mapsto \text{(diagram)} \equiv \begin{bmatrix} \sqrt{1-\vartheta_+}\,e^{i\phi_2} \\ \sqrt{\vartheta_+}\,e^{i\phi_3} \end{bmatrix}. \tag{92b}
$$

Proceeding analogously in the case of the right relations we obtain

$$
\text{(diagram)} \mapsto \text{(diagram)} \equiv \begin{bmatrix} 1 \\ 0 \\ 0 \end{bmatrix}, \qquad\qquad \text{(diagram)} \mapsto \text{(diagram)} \equiv \begin{bmatrix} 1-\vartheta_- \\ \vartheta_- \\ -\vartheta_-^2(1-\vartheta_-)^{-1} \end{bmatrix}, \tag{93a}
$$

$$
\text{(diagram)} \mapsto \text{(diagram)} \equiv \begin{bmatrix} e^{i\phi_1} \\ 0 \end{bmatrix}, \qquad\qquad \text{(diagram)} \mapsto \text{(diagram)} \equiv \begin{bmatrix} \sqrt{1-\vartheta_-}\,e^{i\phi_2} \\ \sqrt{\vartheta_-}\,e^{i\phi_3} \end{bmatrix}. \tag{93b}
$$

This immediately implies that the solution is consistent only if

$$
\vartheta_- = \vartheta_+ = \vartheta, \tag{94}
$$

which means that solvable product states cannot relax to a GGE with an imbalance of particles.

This can be understood by noting that a state $|\tilde{\Psi}_0\rangle$ can relax to a GGE (71) with $\vartheta_- \neq \vartheta_+$ only if

$$
\langle \tilde{\Psi}_0 | N_+ - N_- | \tilde{\Psi}_0 \rangle \neq 0, \tag{95}
$$

where $N_+$ and $N_-$ are reported in Eq. (72). In particular, if the state is invariant under two-site shifts we have

$$
\langle \tilde{\Psi}_0 | N_+ - N_- | \tilde{\Psi}_0 \rangle = \langle \tilde{\Psi}_0 | P_1 P_2 - P_2 P_3 | \tilde{\Psi}_0 \rangle, \tag{96}
$$

which is always zero if $|\tilde{\Psi}_0\rangle$ is a product state. Therefore, to find states that relax to a GGE with $z_- \neq z_+$, one should try with initial states in a more general MPS form. The question of whether there are nontrivial initial MPSs satisfying the set of boundary relations (89,90) is so far still open.

## 4.4 Summary of diagrammatic relations

We conclude this section by summing up the algebraic relations satisfied by the fixed points. In particular, the space transfer matrices

$$
\tilde{\mathbb{W}}_s = \text{(diagram)}, \qquad \tilde{\mathbb{W}} = \text{(diagram)}, \tag{97}
$$

corresponding to the stationary Gibbs state (74) and solvable initial states (92b) and (93b) respectively, exhibit left and right fixed-points that can be represented as MPSs with bond dimension 3 (83)

$$\langle L_{\mathrm s}| = \quad , \qquad |R_{\mathrm s}\rangle = \quad , \qquad \langle L| = \quad , \qquad |R\rangle = \quad . \tag{98}$$

Bulk tensors (given by Eq. (83) and (148)) and boundary vectors (see Eq. (149), (150), (92a) and (93a)) constituting the fixed-points, together with the initial state

$$|\Psi_0\rangle = \underbrace{\qquad\qquad\qquad\qquad}_{2L} , \tag{99}$$

and the stationary MPO (73), satisfy the following set of *local* algebraic relations

$$\tag{100a}$$

$$\tag{100b}$$

$$\tag{100c}$$

$$\tag{100d}$$

## 5   Solution of inhomogeneous quenches from solvable states

Now that we have the explicit form of fixed-points $\langle L|$ and $|R\rangle$ for a class of initial states, let us focus on the dynamics of local observables after the quenches from these states. In particular, we consider the bipartitioning protocol, where at time $t = 0$ the two halves of the chain are prepared in different solvable product states, parametrised respectively by $\vartheta_{\mathrm L}$ and $\vartheta_{\mathrm R}$

$$|\Psi_0\rangle = \left(|\psi_{1,\mathrm L}\rangle \otimes |\psi_{2,\mathrm L}\rangle\right)^{\otimes L/2} \otimes \left(|\psi_{1,\mathrm R}\rangle \otimes |\psi_{2,\mathrm R}\rangle\right)^{\otimes L/2} ,$$
$$|\psi_{1,\mathrm L/R}\rangle = \begin{bmatrix} \mathrm e^{\mathrm i\phi_1} \\ 0 \end{bmatrix}, \qquad |\psi_{2,\mathrm L/R}\rangle = \begin{bmatrix} \mathrm e^{\mathrm i\phi_2}\sqrt{1-\vartheta_{\mathrm L/R}} \\ \mathrm e^{\mathrm i\phi_3}\sqrt{\vartheta_{\mathrm L/R}} \end{bmatrix}. \tag{101}$$

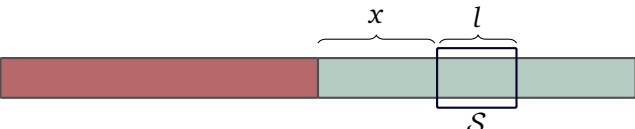

Figure 2: Sketch of the inhomogeneous quench setup considered in Sec. 5. We consider the dynamics after a bipartitioning protocol of a finite subsystem $\mathcal{S}$ of size $l$ is at distance $x$ from the junction.

Note that this more general class of initial states includes quenches from homogeneous solvable initial states ($\vartheta = \vartheta_{\mathrm{L}} = \vartheta_{\mathrm{R}}$), therefore we can focus on initial states (101) without loosing generality.

In our analysis we consider the reduced density matrix of a finite subsystem $\mathcal{S}$, as it encodes expectation values of all local observables. The length of the subsystem $l$ is fixed and we denote its relative position with respect to the junction by $x$, see Fig. 2. Depending on the scaling of the size of the subsystem and its distance from the junction with time, several qualitatively different regimes emerge. Specifically here we investigate two different ones

(i) Both $l$ and $x$ are fixed, i.e. the do not scale with $t$.

(ii) The subsystem size $l$ is fixed but its position scales linearly with time.

Let us address these two regimes separately beginning with Case (i).

## 5.1 Subsystem at a fixed distance from the junction

We consider a subsystem $\mathcal{S}$ of length $l > 0$, with the edges at sites $-x < 0$ and $l - x$ which, for simplicity, we assume to be *even*. In this case the density matrix reduced to $\mathcal{S}$ is represented by the following tensor network

$$\rho_l(t) = \qquad \qquad \qquad \qquad . \qquad (102)$$

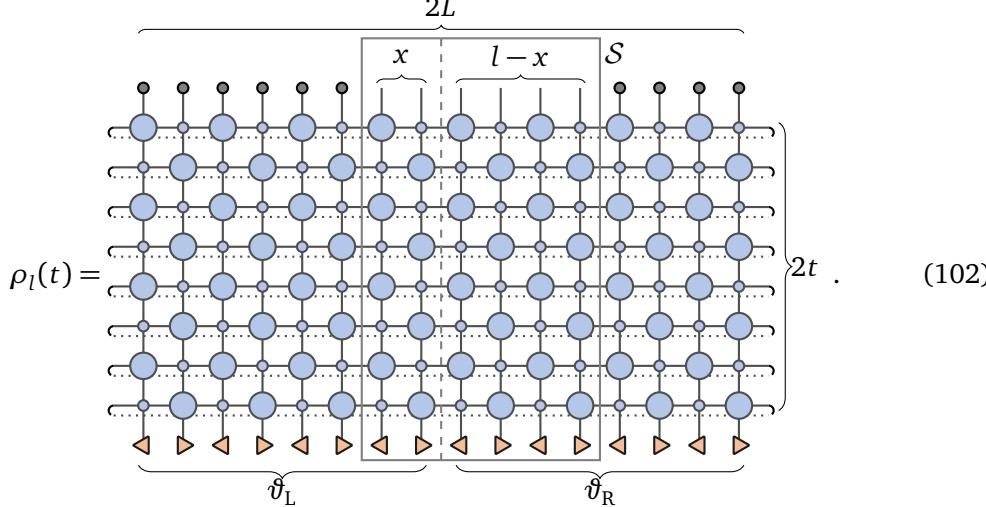

Note that this is the most general setup, as a subsystem with edges at position $x$ and $x + l$ can be always extended to the left so that it contains the junction, and then the additional sites are traced over at the end, which does not modify the argument presented in this section (as long as $x$ is fixed).

We assume that the full system-size $2L$ is strictly larger than $4t$, so that we can recast the

reduced density matrix in terms of left and right fixed-points $\left\langle L_{\vartheta_{\mathrm{L}}}\right|$ and $\left|R_{\vartheta_{\mathrm{R}}}\right\rangle$,

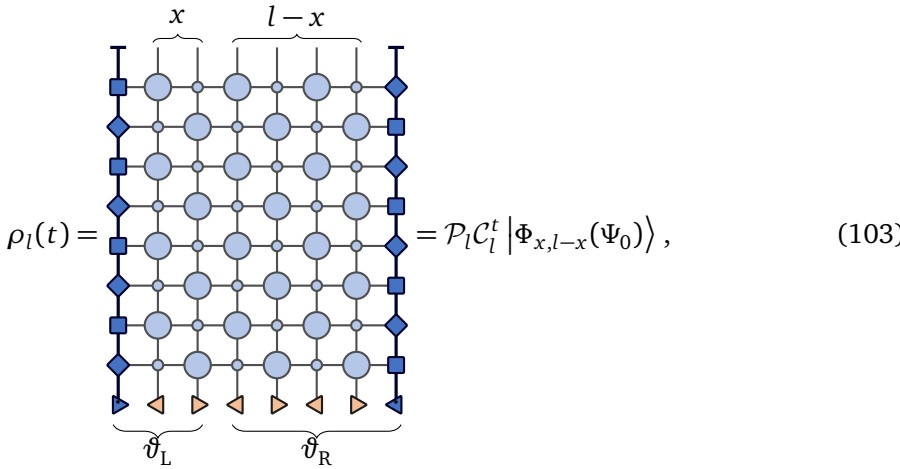

$$\rho_l(t) = \quad = \mathcal{P}_l \mathcal{C}_l^t \left|\Phi_{x,l-x}(\Psi_0)\right\rangle, \tag{103}$$

where we introduced the transfer matrix in the time-direction

$$\mathcal{C}_l = \qquad , \tag{104}$$

the "bottom state"

$$\left|\Phi_{x,y}(\Psi_0)\right\rangle = \qquad , \tag{105}$$

and the projector

$$\mathcal{P}_l = \qquad , \tag{106}$$

which only acts on the auxiliary space. We note that the normalisation factor was omitted from Eq. (103) because the overlap between the two fixed-points is 1 for all $\vartheta_{\mathrm{L}}$ and $\vartheta_{\mathrm{R}}$ (see Appendix B.1).

Eq. (103) implies that the dynamics of the finite subsystem $\mathcal{S}$ is completely specified by a finite dimensional tensor network. The latter can be contracted with a complexity that is exponential in the subsystem size $l$ but *polynomial* in time. However, using the properties of the local tensors, we can characterise the relaxation when $t \to \infty$ of *any* finite size.

We start by observing that the map $\mathcal{C}_l$ has always an eigenvalue one and two of its fixed points are easily expressed in terms of the folded identity operator and the stationary state MPO (73).

**Property 2.** *The MPS $|1_l\rangle$ defined as*

$$|1_l\rangle = \frac{1}{\underbrace{\phantom{xxx}}_{(1+\vartheta_{\mathrm{L}}+\vartheta_{\mathrm{R}})^{-1}}} \qquad , \tag{107}$$

*is a right eigenvector of $\mathcal{C}_l$ corresponding to an eigenvalue 1, i.e.*

$$\mathcal{C}_l \left|1_l\right\rangle = \left|1_l\right\rangle. \tag{108}$$

*Similarly, the left eigenvector $\left\langle\bar{1}_l\right|$ is diagrammatically expressed as*

$$\left\langle\bar{1}_l\right| = \quad\quad\quad\quad\quad\quad\quad\quad\quad\quad \tag{109}$$

   Property 2 can be understood intuitively by noting that both (107) and (109) are stationary when we remove the boundary degrees of freedom and assume periodic boundaries on $l$ sites: in this case $\left\langle\bar{1}_l\right|$ reduces to the folded representation of the identity matrix, while $\left|1_l\right\rangle$ becomes the GGE given by (73). The non trivial aspect is that they can be made stationary also in the presence of a boundary upon choosing appropriate boundary conditions (see Appendix B.2 for details). We also note that by projecting out the auxiliary degrees of freedom, the right eigenvector $\left|1_l\right\rangle$ is mapped directly to the GGE reduced to a finite subsystem of $l$ sites

$$\rho_{\mathrm{GGE},l}(\vartheta_{\mathrm{L}}, \vartheta_{\mathrm{R}}) = \mathcal{P}_l \left|1_l\right\rangle, \tag{110}$$

which is defined as

$$\rho_{\mathrm{GGE},l}(\vartheta_{+}, \vartheta_{-}) = \frac{1}{1 + \vartheta_{+} + \vartheta_{-}} \quad\quad\quad\quad\quad\quad\quad . \tag{111}$$

We are now in a position to show that the reduced density matrix $\rho_l(t)$ relaxes to the state $\rho_{\mathrm{GGE},l}(\vartheta_{\mathrm{L}}, \vartheta_{\mathrm{R}})$ exponentially fast with a finite rate determined by the spectrum of the $9 \times 9$ matrix $\mathcal{C}_0$ (cf. (106)).

**Property 3.** *If $2t > 3x$ and $2t > 3(l-x)$, the reduced density matrix $\rho_l(t)$ is equal to $\rho_{\mathrm{GGE},l}(\vartheta_{\mathrm{L}}, \vartheta_{\mathrm{R}})$ up to exponentially small corrections. Explicitly*

$$\rho_l(t) = \rho_{\mathrm{GGE},l}(\vartheta_{\mathrm{L}}, \vartheta_{\mathrm{R}}) + \mathcal{O}\left(\Lambda_1^{t - \frac{3}{2}\max\{x, l-x\}}\right), \tag{112}$$

*where*

$$\Lambda_1 = -\frac{\vartheta_{\mathrm{L}} + \vartheta_{\mathrm{R}} - \vartheta_{\mathrm{L}}\vartheta_{\mathrm{R}}}{2}\left(1 + \sqrt{1 - \frac{4\vartheta_{\mathrm{L}}\vartheta_{\mathrm{R}}}{\vartheta_{\mathrm{L}} + \vartheta_{\mathrm{R}} - \vartheta_{\mathrm{L}}\vartheta_{\mathrm{R}}}}\right), \tag{113}$$

*is the largest subleading eigenvalue of $\mathcal{C}_0$.*

*Proof.* The idea is to use the "zipping conditions" (100a), (100b) and (100d) to simplify the diagram (103) by absorbing initial states and time-evolution tensors at the boundaries. In particular, if $2t > 3m$, with $m = \max\{x, l-x\}$, all the dependence on the initial state can be

absorbed into fixed-points, as illustrated by the following diagram

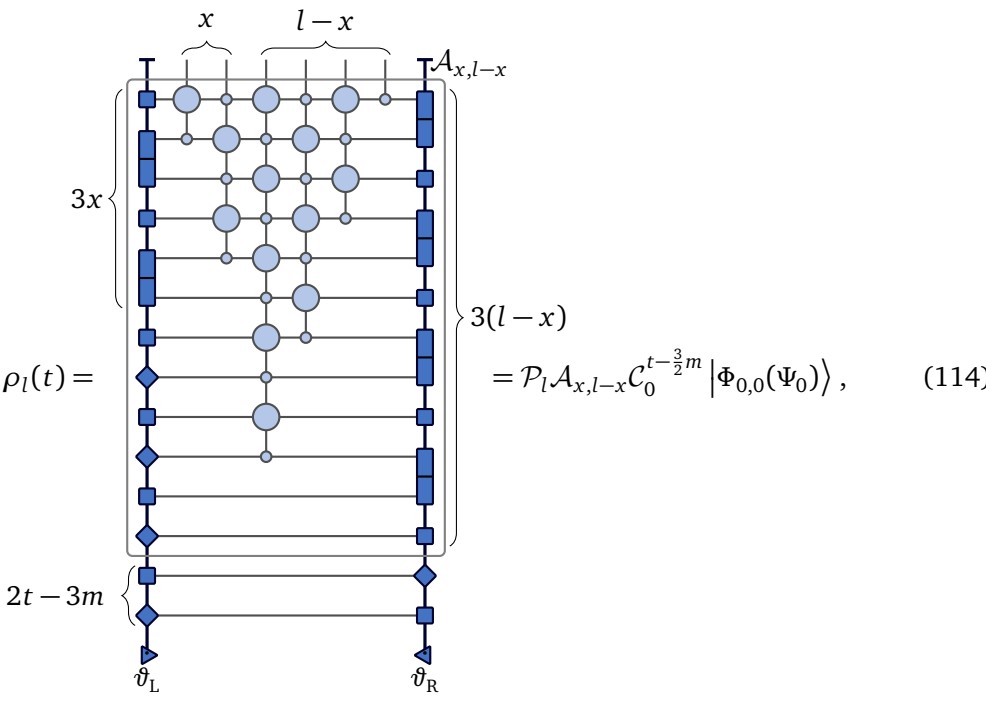

$$\rho_l(t) = \qquad\qquad = \mathcal{P}_l \mathcal{A}_{x,l-x} \mathcal{C}_0^{t-\frac{3}{2}m} \left|\Phi_{0,0}(\Psi_0)\right\rangle, \qquad (114)$$

where we denoted the top part of the tensor network by $\mathcal{A}_{x,l-x}$.

Since $\mathcal{A}_{x,l-x}$ does not scale with time, the long-time behaviour of the above diagram is determined by the spectral properties of $\mathcal{C}_0$. In particular, by explicitly diagonalising the $9 \times 9$ matrix (see Appendix B.3), one can straightforwardly verify that it has 3 non-zero eigenvalues with geometric and algebraic multiplicity 1. These are 1, $\Lambda_1$, and

$$\Lambda_2 = -\frac{\vartheta_L + \vartheta_R - \vartheta_L \vartheta_R}{2}\left(1 - \sqrt{1 - \frac{4\vartheta_L \vartheta_R}{\vartheta_L + \vartheta_R - \vartheta_L \vartheta_R}}\right). \qquad (115)$$

Moreover, the bottom state in Eq. (105) can be expressed in terms of the corresponding eigenvectors as

$$\left|\Phi_0(\Psi)\right\rangle = \left|1_0\right\rangle + \gamma_1 \left|\Lambda_{1,0}\right\rangle + \gamma_2 \left|\Lambda_{2,0}\right\rangle, \qquad (116)$$

with precise values of $\gamma_{1,2}$ specified in (165). The full reduced density matrix can be therefore expressed as the sum of all three contributions,

$$\rho_l(t) = \mathcal{P}_l \mathcal{A}_{x,l-x}\left|1_0\right\rangle + \Lambda_1^{t-\frac{3}{2}m}\gamma_1 \mathcal{P}_l \mathcal{A}_{x,l-x}\left|\Lambda_{1,0}\right\rangle + \Lambda_2^{t-\frac{3}{2}m}\gamma_2 \mathcal{P}_l \mathcal{A}_{x,l-x}\left|\Lambda_{2,0}\right\rangle. \qquad (117)$$

Now we note that the second and third term are exponentially suppressed, as $\mathcal{A}_{x,l-x}$ does not change with time and only depends on the size of the system $l$ and the distance from the junction $x$, which are both fixed. The dominant term is therefore $\mathcal{A}_{x,l-x}\left|1_0\right\rangle$ and it can be again simplified, by repeatedly using the algebraic relations leading from (103) to (114) "backwards", only now the boundary relations used at the bottom are the ones involving the

stationary MPS (100c) rather than (100d). Explicitly, we obtain

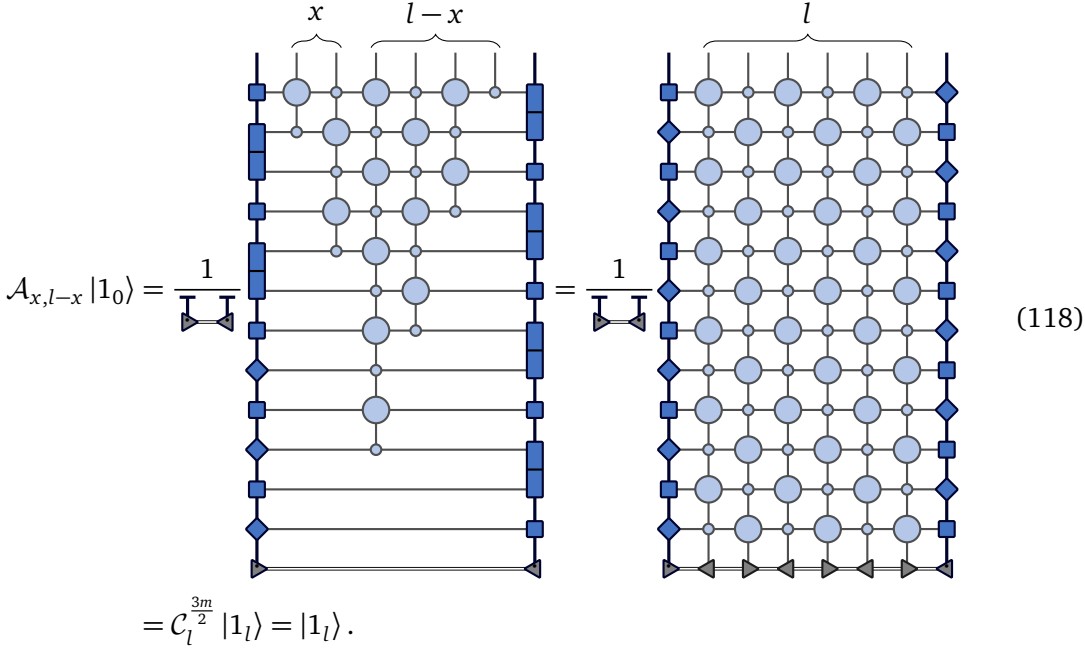

$$\mathcal{A}_{x,l-x}\,|1_0\rangle = \frac{1}{\bowtie} \qquad = \frac{1}{\bowtie} \qquad \tag{118}$$

$$= \mathcal{C}_l^{\frac{3m}{2}}\,|1_l\rangle = |1_l\rangle\,.$$

From here the proof of (112) follows immediately

$$\rho_l(t) = \mathcal{P}_l\,|1_l\rangle + \mathcal{O}\left(|\Lambda_1|^{t-\frac{3}{2}\max\{x,l-x\}}\right) = \rho_{\mathrm{GGE},l}(\vartheta_{\mathrm{L}},\vartheta_{\mathrm{R}}) + \mathcal{O}\left(|\Lambda_1|^{t-\frac{3}{2}\max\{x,l-x\}}\right)\,. \tag{119}$$

$$\square$$

Property 3 implies that *any* local observable $\mathcal{O}_{[-x,-x+l]}$, supported on the section of the lattice between sites $-x$ and $-x + l$ (see (102)), relaxes to the GGE value with a finite rate

$$\langle\Psi_t|\mathcal{O}_{[-x,-x+l]}|\Psi_t\rangle - \mathrm{tr}\left(\rho_{\mathrm{GGE}}\mathcal{O}_{[-x,-x+l]}\right) \propto \Lambda_1^{t-\frac{3}{2}\max\{x,l-x\}}\,. \tag{120}$$

The above result also describes quenches from *homogeneous* solvable states, if we take $\vartheta_{\mathrm{L}} = \vartheta_{\mathrm{R}} = \vartheta$. In this case the reduced density matrix relaxes to the Gibbs state

$$\rho_{\mathrm{GE},l}(\vartheta) = \rho_{\mathrm{GGE},l}(\vartheta,\vartheta)\,, \tag{121}$$

with the rate $\Lambda_1 = \Lambda_1|_{\vartheta_{\mathrm{L}}=\vartheta_{\mathrm{R}}=\vartheta}$,

$$\langle\Psi_t|\mathcal{O}_{[0,l]}|\Psi_t\rangle - \mathrm{tr}\left(\rho_{\mathrm{GE}}\mathcal{O}_{[0,l]}\right) \propto \Lambda_1^{t-\frac{3}{4}l}\,. \tag{122}$$

Before moving to Case (ii) let us briefly comment on the physics of our result. Indeed, since the exponential relaxation of all local observables is a feature typically associated with chaotic systems, it can be surprising to see a result like Eq. (120) for an integrable model. Even though the lack of other solvable examples of interacting integrable dynamics makes it hard to have a comprehensive discussion, we can compare our results with the picture emerging from a systematic study of the free case, see e.g. [103]. In free systems the expectation value of local observables relax either in exponential or power-law fashion, depending on the specific observable. In particular, operators that are *local* with respect to elementary excitations relax algebraically, while the *nonlocal* ones relax exponentially. This picture is believed to carry over to the interacting case, but so far it has been tested only in a handful of examples. For instance, Ref. [104] used a form-factor expansion to show that a particular observable (nonlocal w.r.t.

the elementary excitations) relaxes exponentially after a quench in the sine-Gordon model, while Ref. [105] used a hybrid analytic-numerical method to show that "generic" observables in the one-dimensional Bose gas relax in power-law fashion.

To summarise, the behaviour described in Eq. (120) is indeed special, because in a generic integrable model one expects to see both exponential and power law relaxation. This is ultimately due to the simple structure of the fixed point states. Moreover, as we argue later (see Section 6), the exponential decay of all local observables described by (120) can also be understood from the hydrodynamic point of view. Indeed, quasiparticles in Rule 54 have both a *maximal* velocity (which is a consequence of the local evolution) and a *minimal* one (which is a consequence of the precise rules of the dynamics, see [57]). This excludes the possibility of power-law relaxation for systems at finite distance from the junction.

## 5.2 Position of the subsystem scales with time

Let us now consider the situation in which the subsystem $\mathcal{S}$ is at a distance from the junction that scales linearly with time. Namely

$$x = \zeta t + x_0, \tag{123}$$

with $x_0 = \mathcal{O}(t^0)$. In this case the reduced density matrix is represented as

$$\rho_l(x,t) = \qquad\qquad\qquad\qquad\qquad\qquad\qquad . \tag{124}$$

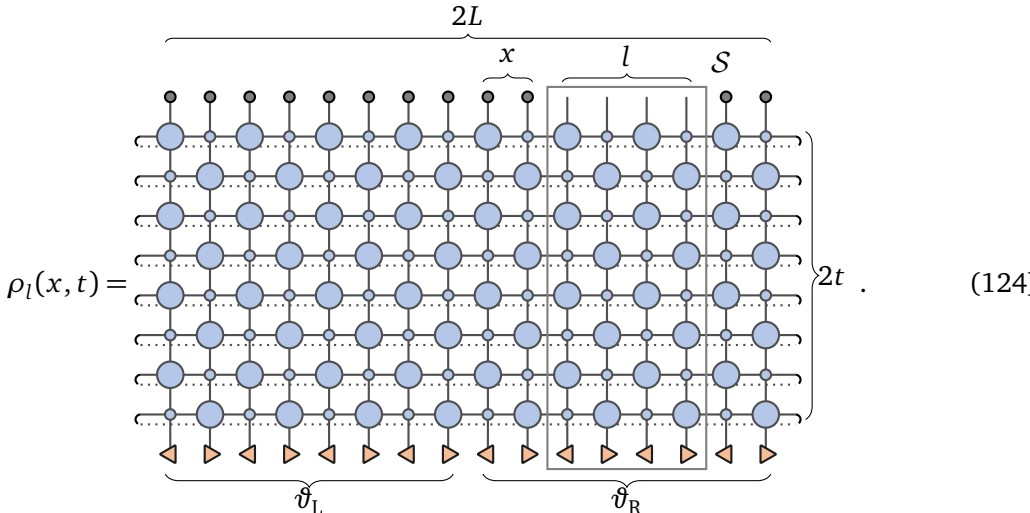

Moreover, let us also introduce the symbol $\rho_{l,\zeta}$ to denote the reduced density matrix in the scaling limit of infinite time and distance, i.e.

$$\rho_{l,\zeta} = \lim_{\substack{|x|,t\to\infty \\ x/t=\zeta}} \rho_l(x,t). \tag{125}$$

In the thermodynamic limit the reduced density matrix is given by

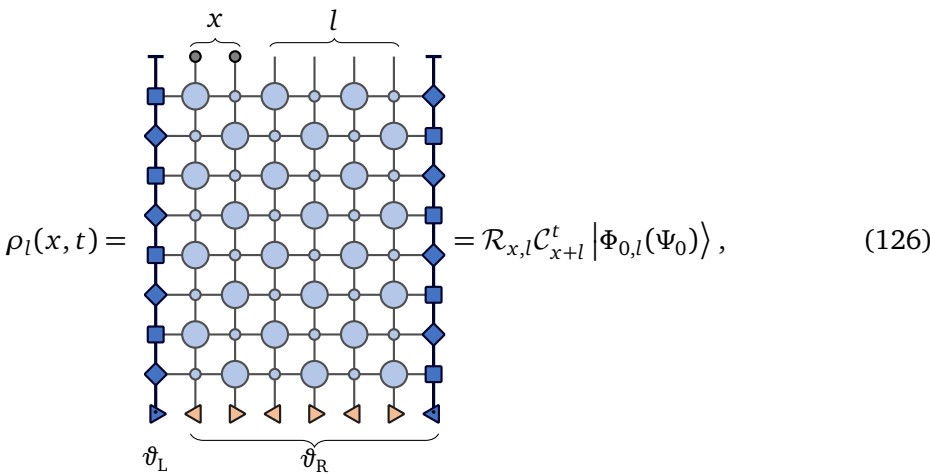

$$\rho_l(x,t) = \qquad\qquad = \mathcal{R}_{x,l}\mathcal{C}^t_{x+l}\left|\Phi_{0,l}(\Psi_0)\right\rangle, \qquad (126)$$

where $\mathcal{C}_l$ and $\left|\Phi_{x,y}\right\rangle$ are defined as before (cf. (106)), while $\mathcal{R}_{x,l}$ projects out the auxiliary degrees of freedom and the first $x$ physical sites, i.e.

$$\mathcal{R}_{x,l} = \qquad\qquad . \qquad (127)$$

As we are now scaling the position $x$ with time, the width of the tensor network is not constant and we are not directly able to contract it using the results of the previous subsection. Nonetheless, there are two regimes for which we can find the exact steady state. The first one is $|\zeta| > 2$ and corresponds to a subsystem positioned outside of the causal light-cone, which relaxes as if the initial state was homogeneous. The second regime corresponds to $|\zeta| < 2/3$. Let us describe these two regimes starting from the former.

### 5.2.1 Out of the lightcone: $|\zeta| \geq 2$

For $|x|/t = |\zeta| > 2$ we can simplify Eq. (126) by using the local structure of the time evolution. Indeed, we have the identity

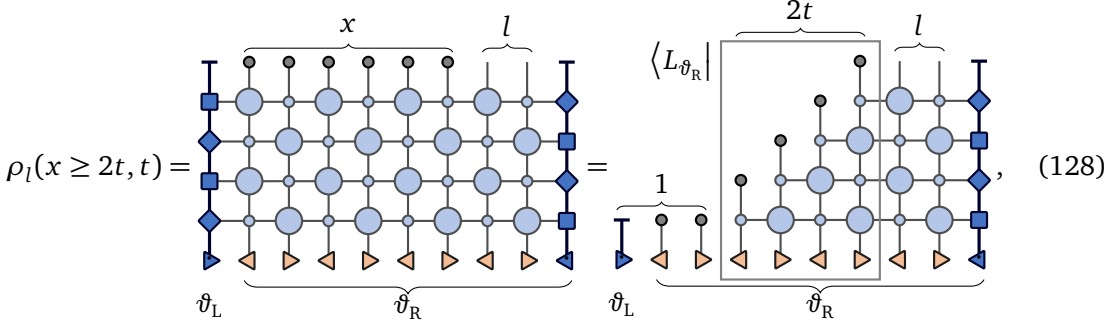

$$\rho_l(x \geq 2t, t) = \qquad\qquad = \qquad\qquad , \qquad (128)$$

which follows from the unitarity of the time-evolution and is proven using the local algebraic relations (160) (which we used in the proof of the second part of Property 2, see Appendix B.2). After noting that the triangularly shaped part of the tensor network (in the grey box) is precisely the left fixed-point corresponding to the parameter on the right $\vartheta_{\rm R}$ (cf. (54a)), we obtain

exactly the homogeneous limit of the diagram (103)

$$\rho_l(x \geq 2t, t) = \quad 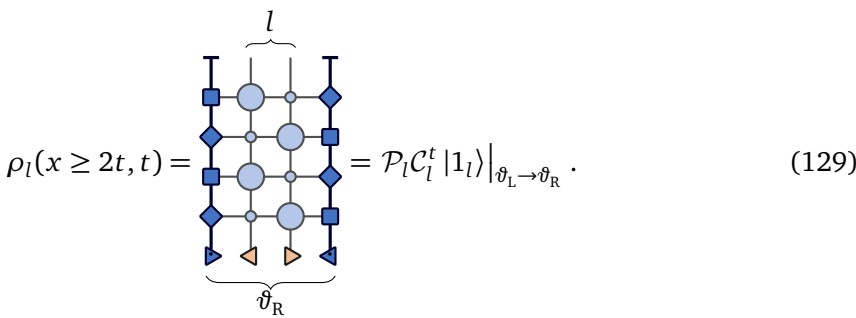 \quad = \mathcal{P}_l \mathcal{C}_l^t \left|1_l\right\rangle\big|_{\vartheta_{\mathrm{L}} \to \vartheta_{\mathrm{R}}} . \tag{129}$$

Since there is no explicit dependence on $x$, we can immediately take the limit $x, t \to \infty$. In particular, in analogy with the situation considered in Section 5.1, we obtain

$$\rho_{l,\zeta \geq 2} = \rho_{\mathrm{GGE},l}(\vartheta_{\mathrm{R}}, \vartheta_{\mathrm{R}}) = \rho_{\mathrm{GE},l}(\vartheta_{\mathrm{R}}). \tag{130}$$

A completely analogous reasoning gives

$$\rho_{l,\zeta \leq -2} = \rho_{\mathrm{GE},l}(\vartheta_{\mathrm{L}}). \tag{131}$$

### 5.2.2  Close to the junction: $|\zeta| < 2/3$

In this regime, we can apply an argument analogous to the one used to prove Property 3. We start by assuming

$$2t > 3(x + l), \tag{132}$$

which enables us to reduce the diagram (126) to a form analogous to (117)

$$\begin{aligned}
\rho_l(x, t) &= \mathcal{R}_{x,l} \left|1_{l+x}\right\rangle + \gamma_1 \Lambda_1^{t-\frac{3}{2}(x+l)} \mathcal{R}_{x,l} \mathcal{A}_{0,l+x} \left|\Lambda_{1,0}\right\rangle + \gamma_2 \Lambda_2^{t-\frac{3}{2}(x+l)} \mathcal{R}_{x,l} \mathcal{A}_{0,l+x} \left|\Lambda_{2,0}\right\rangle \\
&= \rho_{\mathrm{GGE},l}(\vartheta_{\mathrm{L}}, \vartheta_{\mathrm{R}}) + \gamma_1 \Lambda_1^t \mathcal{R}_{x,l} \left|\Lambda_{1,(0,x+l)}\right\rangle + \gamma_2 \Lambda_2^t \mathcal{R}_{x,l} \left|\Lambda_{2,(0,x+l)}\right\rangle ,
\end{aligned} \tag{133}$$

where we introduced the following notation for the subleading eigenvectors of $\mathcal{C}_{l+x}$

$$\left|\Lambda_{1/2,(0,x+l)}\right\rangle = \Lambda_{1/2}^{-\frac{3}{2}(x+l)} \mathcal{A}_{0,x+l} \left|\Lambda_{1/2,0}\right\rangle , \tag{134}$$

and we took into account

$$\mathcal{R}_{x,l} \left|1_{x+l}\right\rangle = \mathcal{P}_l \left|1_l\right\rangle = \rho_{\mathrm{GGE},l}(\vartheta_{\mathrm{L}}, \vartheta_{\mathrm{R}}), \tag{135}$$

which follows from $\langle L_{\mathrm{s}}| W_{\mathrm{s}} = \langle L_{\mathrm{s}}|$. The main difference with respect to the case described by Property 3 is that in the scaling regime $\mathcal{A}_{0,x+l}$ *grows* with $t$, therefore one has to explicitly verify that $\left\|\left|\Lambda_{1,(0,x+l)}\right\rangle\right\|$ and $\left\|\left|\Lambda_{2,(0,x+l)}\right\rangle\right\|$ can be bounded *independently* of $x$. As is shown in Appendix B.3, this is indeed the case. Therefore, the subleading terms in (133) are again exponentially suppressed and the reduced density matrix in the scaling regime for $\zeta < 2/3$ (cf. (132)) relaxes to the GGE

$$\rho_{l,|\zeta| < \frac{2}{3}} = \rho_{\mathrm{GGE},l}(\vartheta_{\mathrm{L}}, \vartheta_{\mathrm{R}}). \tag{136}$$

# 6 Comparison with GHD

The scaling regime of the inhomogeneous quench considered in the previous section is the setup in which GHD applies most directly [11, 58, 59]. This means that our exact results can be used to provide a, hitherto missing, independent verification of the GHD predictions. Indeed, up to now, the only predictions of GHD that have been verified by an independent analytical calculation are those concerning dynamical correlations on homogeneous equilibrium states [106, 107], which were recovered in Ref. [108] in the context of a strong coupling expansion. Instead, only partial results [109, 110] are currently available for inhomogeneous quench problems.

Since GHD is expressed in the language of Thermodynamic Bethe Ansatz (TBA) [111, 112] we start by reporting some basic facts about the TBA description of Rule 54 (for further details see [57] and the supplemental material of Ref. [100]). The basic premise of TBA is that, in the thermodynamic limit, expectation values of local observables on eigenstates only depend on some gross *macroscopic* properties of the eigenstates. In particular for Rule 54 these macroscopic properties are the densities $n_+$ and $n_-$ of right and left moving particles. One then considers *macrostates* formed by collections of microscopic eigenstates of the time evolution operator with the same densities. In particular, a combinatorial calculation reveals that a given macrostate corresponds to $N_L \simeq e^{Ls[n_+, n_-]}$ eigenstates of the Hamiltonian. Here we introduced the Yang-Yang entropy

$$s[n_+, n_-] = \sum_{\nu \in \{\pm\}} \frac{n_\nu}{n_{t,\nu}} \log \frac{n_\nu}{n_{t,\nu}} + \left(1 - \frac{n_\nu}{n_{t,\nu}}\right) \log \left(1 - \frac{n_\nu}{n_{t,\nu}}\right), \tag{137}$$

and the density of *slots* that can be filled by particles

$$n_{t,\nu} = 1 - n_\nu + n_{-\nu}, \quad \nu = \pm. \tag{138}$$

The physical meaning of the above equation is that, because of the interactions, the density of available slots that particles can occupy depends on $n_+$ and $n_-$.

One can make elementary excitations on the macrostate $\{n_+, n_-\}$ by adding a left/right moving particle. It turns out that [113], because of the interactions, the velocity of this excitation is not $\pm 2$, as it would be in the vacuum, but it gets renormalised to [57, 100]

$$v_\nu = 2\nu \left(1 - \frac{2n_{-\nu}}{1 + n_\nu + n_{-\nu}}\right). \tag{139}$$

An interesting macrostate is the one corresponding to the microcanonical representation of the GGE (71). The latter is specified by densities $\{n_+, n_-\}$ fulfilling

$$\epsilon_\nu = \mu_\nu + \log \frac{1 + e^{-\epsilon_\nu}}{1 + e^{-\epsilon_{-\nu}}}, \qquad \frac{n_{t,\nu} - n_\nu}{n_\nu} = e^{\epsilon_\nu}. \tag{140}$$

Here there are two important things to note. First, exponentiating these relations and comparing with (87) one directly verifies

$$\vartheta_\pm = \frac{n_\pm}{n_{t,\pm}}. \tag{141}$$

Namely $\vartheta_\pm$ (cf. (84) and (86)) are nothing but the filling functions of the GGE (71). Second, since by varying $(\mu_+, \mu_-) \in \mathbb{R}^2$ we can reproduce all $(n_+, n_-) \in [0,1]^2$, every macrostate can be thought of as a microcanonical representation of a GGE (71). Note that in the case $\mu_+ = \mu_-$ the expressions become

$$n_\nu = \frac{1}{1 + e^\mu}, \qquad n_{t,\nu} = n_t = 1, \qquad \vartheta_\nu = \vartheta = \frac{1}{1 + e^\mu}, \qquad v_\nu = 2\nu \frac{1 + e^\mu}{3 + e^\mu}. \tag{142}$$

$$\rho_{\mathrm{GE},l}(\vartheta_{\mathrm{L}}) \qquad \rho_{\mathrm{GGE},l}(\vartheta_{\mathrm{L}},\vartheta_{\mathrm{R}}) \qquad \rho_{\mathrm{GE},l}(\vartheta_{\mathrm{R}})$$

$$-2 \qquad -\tfrac{2}{3} \quad 0 \quad \tfrac{2}{3} \qquad 2 \qquad \zeta$$

Figure 3: Summary of the results in the scaling limit. For $|\zeta| > 2$ and $|\zeta| < \frac{2}{3}$ (rectangles with darker colours) we independently prove the GHD prediction, while for $\frac{2}{3} < |\zeta| < 2$ (light rectangles) the microscopic verification is still missing.

Now we have all the ingredients to find a description of the inhomogeneous quench. We assume that at large time $t$, the state at the position $x$ can be locally approximated by a GGE that depends on the ray $\zeta = x/t$: we describe it by two ray-dependent filling functions $\vartheta_{\pm,\zeta}$. The two limiting values for $\zeta \to \pm\infty$ are given by the stationary states to which the initial states on the left and right halves of the chain relax

$$\lim_{\zeta\to-\infty} \vartheta_{\nu,\zeta} = \vartheta_{\mathrm{L}}, \qquad \lim_{\zeta\to\infty} \vartheta_{\nu,\zeta} = \vartheta_{\mathrm{R}}, \qquad \forall\, \nu \in \{+,-\}, \tag{143}$$

while GHD predicts [11, 58, 59] that the state for an intermediate value of $\zeta$ is given by

$$\vartheta_{\nu,\zeta} = \begin{cases} \vartheta_{\mathrm{L}}, & \nu_\nu(\zeta) < \zeta \\ \vartheta_{\mathrm{R}}, & \nu_\nu(\zeta) > \zeta \end{cases}, \tag{144}$$

where

$$\nu_\nu(\zeta) = \frac{2\nu}{1 + 2\vartheta_{-\nu,\zeta}}. \tag{145}$$

In our case the above relations can be solved exactly and yield

$$\vartheta_{+,\zeta} = \begin{cases} \vartheta_{\mathrm{L}}, & \zeta < \frac{2}{1+2\vartheta_{\mathrm{R}}}, \\ \vartheta_{\mathrm{R}}, & \zeta > \frac{2}{1+2\vartheta_{\mathrm{R}}}, \end{cases} \qquad \vartheta_{-,\zeta} = \begin{cases} \vartheta_{\mathrm{L}}, & \zeta < -\frac{2}{1+2\vartheta_{\mathrm{L}}}, \\ \vartheta_{\mathrm{R}}, & \zeta > -\frac{2}{1+2\vartheta_{\mathrm{L}}}. \end{cases} \tag{146}$$

This implies that the hydrodynamic prediction for the reduced density matrix in the scaling regime is

$$\rho_{l,\zeta} = \begin{cases} \rho_{\mathrm{GE},l}(\vartheta_{\mathrm{L}}), & \zeta < -\frac{2}{1+2\vartheta_{\mathrm{L}}}, \\ \rho_{\mathrm{GGE},l}(\vartheta_{\mathrm{L}},\vartheta_{\mathrm{R}}), & -\frac{2}{1+2\vartheta_{\mathrm{L}}} < \zeta < \frac{2}{1+2\vartheta_{\mathrm{R}}}, \\ \rho_{\mathrm{GE},l}(\vartheta_{\mathrm{R}}), & \zeta > \frac{2}{1+2\vartheta_{\mathrm{R}}}. \end{cases} \tag{147}$$

As is graphically summarised in Fig. 3 the GHD prediction (147) agrees with our exact results in all regimes that we can access. Indeed, the result (136) implies the relaxation to $\rho_{\mathrm{GGE},l}(\vartheta_L, \vartheta_R)$ for $|\zeta| < 2/3$, which is always contained inside the interval $-2/(1+2\vartheta_L) < \zeta < 2/(1+2\vartheta_R)$. Similarly, $2 > 2/(1+2\vartheta_R)$ and $-2 < -2/(1+2\vartheta_L)$, which means that Eqs. (130) and (131) are compatible with the GHD prediction. To the best of our knowledge, this is the first *ab initio* derivation of the GHD prediction for an inhomogeneous quench in an interacting system.

Note that for the intermediate values of the scaling ratio, $2/3 < |\zeta| < 2$, we are not able to directly contract the tensor network (126), and the question of whether or not our approach can be extended to provide a rigorous confirmation of the result (147) over the whole light cone remains open.

# 7 Conclusions

In this paper we studied the out-of-equilibrium dynamics of the quantum cellular automaton Rule 54 using a time-channel approach. We introduced a class of "solvable" initial states for which we could provide an explicit construction of the fixed-points of the space transfer matrix. We used the latter to express the time-evolution of all finite subsystems in terms of finite-dimensional quantum maps and, in turn, to characterise exactly their relaxation. For the class of initial states considered, we showed that *all* local observables relax exponentially fast to Gibbs states. Furthermore, we considered quenches from piecewise-homogeneous states built from solvable initial states, and we demonstrated that they relax to *non-equilibrium* stationary states whose properties are described by the GGEs with two chemical potentials (corresponding to left and right movers). In the accessible regimes, our results agree with the predictions of GHD providing the first independent confirmation of the latter for an inhomogeneous quench in a simple yet interacting many-body system.

The work presented here opens many exciting directions for future research. In particular we can envisage three broad classes of questions.

First, even though our results pertain to an integrable system, our approach did not explicitly rely on integrability. An immediate direction is then to connect our results with the Bethe-Ansatz-based programme for studying the time channel dynamics put forward in Refs. [68–70]. In particular, it is interesting to ask whether the solvable initial states found here correspond to the Bethe-Ansatz "integrable" ones of Ref. [114] (we believe that this is the case because they only produce pairs of quasiparticles [60]) and if a Bethe-Ansatz approach can explain the simple form of the fixed points.

The second set of questions concerns a quantitative characterisation of the effect of conservation laws on the finite-time dynamics. Indeed, even though Rule 54 exhibits exponentially many (in the volume) local integrals of motion [100, 115], the class of states that we studied here relaxes to GGEs depending on only *two* of them. This means in particular that the hydrodynamic regime discussed here effectively involves only two independent continuity equations. It would be very interesting to describe the dynamics (and the eventual hydrodynamic regime) ensuing from states involving increasingly many conservation laws, as it could reveal potential qualitative effects of conservation laws in the finite-time dynamics of local observables. This direction seems within the scope of our approach: one would only need to find fixed points corresponding to more complicated GGEs. We expect these fixed points to maintain an MPS form with a bond dimension corresponding to the one of the MPO representing the stationary state — in the case considered here they are both equal to three.

Finally, and perhaps more interestingly, it is natural to wonder whether our results can be generalised to other systems. The diagrammatic language employed here is largely model-independent, and the algebraic relations summarised in Section 4.4 provide a convenient starting point. It would be interesting to understand whether they can be systematically solved for more general time-evolution tensors and what are precisely the properties that they have to satisfy to exhibit a simple fixed-point solution. Furthermore, one could also think of developing a numerical scheme to find approximations to the fixed-points and hence gaining insights into the relaxation dynamics of a broader class of models (potentially non-integrable).

## Acknowledgements

We thank Lorenzo Piroli for collaboration on closely related projects and useful comments on the draft. We also thank Referee 1 of Paper II for suggesting us to discuss the existing time-channel approaches based on integrability.

**Funding information** This work has been supported by the EPSRC through the grant EP/S020527/1 (KK) and by the Royal Society through the University Research Fellowship No. 201101 (BB).

## A Explicit form of fixed-point bulk and boundary tensors

The two-site fixed-point tensors, which together with tensors (83) satisfy the set of local algebraic relations (100), take the following form,

$$
\begin{aligned}
&\begin{array}{l}00\\00\end{array}_{\vartheta}=\begin{bmatrix}(1-\vartheta)^2 & (1-\vartheta)^2 & -(1-\vartheta)^2\\ (1-\vartheta)\vartheta & (1-\vartheta)\vartheta & -(1-\vartheta)\vartheta\\ (1-\vartheta)\vartheta & -\vartheta^2 & \vartheta^2\end{bmatrix}, &&
\begin{array}{l}00\\10\end{array}_{\vartheta}=\begin{array}{l}00\\01\end{array}_{\vartheta}=\begin{bmatrix}0 & 0 & 0\\ 0 & \vartheta^2 & (1-\vartheta)\vartheta\\ 0 & \vartheta^2 & (1-\vartheta)\vartheta\end{bmatrix},\\[18pt]
&\begin{array}{l}00\\11\end{array}_{\vartheta}=\begin{bmatrix}0 & (1-\vartheta)^2 & -(1-\vartheta)^2\\ 0 & (1-\vartheta)\vartheta & -(1-\vartheta)\vartheta\\ 0 & -\vartheta^2 & \vartheta^2\end{bmatrix}, &&
\begin{array}{l}01\\01\end{array}_{\vartheta}=\begin{array}{l}10\\10\end{array}_{\vartheta}=\begin{bmatrix}\vartheta & 0 & 0\\ 0 & \vartheta & 1-\vartheta\\ 0 & 0 & 0\end{bmatrix},\\[18pt]
&\begin{array}{l}01\\10\end{array}_{\vartheta}=\begin{array}{l}10\\01\end{array}_{\vartheta}=\begin{bmatrix}0 & 1-\vartheta & 0\\ 0 & \vartheta & 0\\ 0 & 0 & \vartheta\end{bmatrix}, &&
\begin{array}{l}11\\00\end{array}_{\vartheta}=\begin{bmatrix}\vartheta & 0 & 0\\ 0 & 0 & 0\\ 0 & 0 & 0\end{bmatrix}, \qquad (148)\\[18pt]
&\begin{array}{l}11\\10\end{array}_{\vartheta}=\begin{array}{l}11\\01\end{array}_{\vartheta}=\begin{bmatrix}0 & 0 & 0\\ (1-\vartheta)\vartheta & 0 & 0\\ (1-\vartheta)\vartheta & 0 & 0\end{bmatrix}, &&
\begin{array}{l}11\\11\end{array}_{\vartheta}=\begin{bmatrix}\vartheta & 0 & 0\\ 1-\vartheta & 0 & 0\\ 0 & 0 & 0\end{bmatrix},\\[18pt]
&\begin{array}{l}01\\11\end{array}_{\vartheta}=\begin{array}{l}01\\00\end{array}_{\vartheta}=0, &&
\begin{array}{l}s_1 r_1\\ s_2 r_2\end{array}_{\vartheta}=\begin{array}{l}r_1 s_1\\ r_2 s_2\end{array}_{\vartheta}.
\end{aligned}
$$

To completely specify the left fixed-point corresponding to the stationary state, we also need the following set of boundary tensors

$$
\begin{aligned}
&{}^{1}\!\blacktriangleright = \begin{bmatrix}1\\0\\0\end{bmatrix}^T, &&
{}^{2}\!\blacktriangleright = \begin{bmatrix}0\\ \frac{\vartheta_+(1-\vartheta_-)}{1-\vartheta_+}\\ \frac{(1-\vartheta_+)\vartheta_-}{1-\vartheta_-}\end{bmatrix}^T, &&
{}^{3}\!\blacktriangleright = \begin{bmatrix}0\\ \frac{\vartheta_+(1-\vartheta_-)}{1-\vartheta_+}\\ -\frac{\vartheta_+\vartheta_-}{1-\vartheta_-}\end{bmatrix}^T,\\[18pt]
&{}^{1}\!\blacktriangleleft = \begin{bmatrix}1-\vartheta_+\\0\\0\end{bmatrix}^T, &&
{}^{2}\!\blacktriangleleft = \begin{bmatrix}0\\ \frac{(1-\vartheta_+)^2\vartheta_-}{1-\vartheta_-}\\ (1-\vartheta_+)\vartheta_-\end{bmatrix}^T, &&
{}^{3}\!\blacktriangleleft = \begin{bmatrix}0\\ -\frac{(1-\vartheta_+)\vartheta_+\vartheta_-}{1-\vartheta_-}\\ \vartheta_+(1-\vartheta_-)\end{bmatrix}^T,
\end{aligned} \qquad (149)
$$

and the equivalent set for the right fixed point is

$$
\begin{aligned}
&{}^{1}\!\blacktriangleright = \begin{bmatrix}1\\ \frac{\vartheta_+(1-\vartheta_-)}{(1-\vartheta_+)^2}\\ 1\end{bmatrix}, &&
{}^{2}\!\blacktriangleright = \begin{bmatrix}\frac{\vartheta_+}{1-\vartheta_+}\\ \frac{1-\vartheta_-}{1-\vartheta_+}\\ \frac{\vartheta_-}{1-\vartheta_-}\end{bmatrix}, &&
{}^{3}\!\blacktriangleright = \begin{bmatrix}0\\ \frac{1-\vartheta_+-\vartheta_-}{1-\vartheta_+}\\ \frac{\vartheta_-(1-\vartheta_+-\vartheta_-)}{(1-\vartheta_-)^2}\end{bmatrix},\\[18pt]
&{}^{1}\!\blacktriangleleft = \begin{bmatrix}1-\vartheta_-\\ \frac{(1-\vartheta_+)\vartheta_-}{1-\vartheta_-}\\ 1-\vartheta_-\end{bmatrix}, &&
{}^{2}\!\blacktriangleleft = \begin{bmatrix}\vartheta_-\\ 1-\vartheta_+\\ \frac{\vartheta_+(1-\vartheta_-)}{1-\vartheta_+}\end{bmatrix}, &&
{}^{3}\!\blacktriangleleft = \begin{bmatrix}\frac{\vartheta_-(1-\vartheta_+-\vartheta_-)}{1-\vartheta_-}\\ 0\\ 0\end{bmatrix}.
\end{aligned} \qquad (150)
$$

# B   Properties of fixed-point tensors

## B.1   Normalisation of the fixed-point MPS

**Property 4.** *The overlap between left and right fixed-points corresponding to (possibly different) solvable initial states is* 1,

$$\langle L_{\vartheta_1} | R_{\vartheta_2} \rangle = \underbrace{\qquad}_{2t} = 1, \tag{151}$$

*independently of the time t and the choice of parameters $\vartheta_1$, $\vartheta_2$.*

*Proof.* The proof follows from the observation that the two-site state at the top of the diagram (151),

$$\top\;\top, \tag{152}$$

is a fixed-point of both

$$\blacksquare\!\!-\!\!\blacklozenge, \qquad \text{and} \qquad \blacklozenge\!\!-\!\!\blacksquare. \tag{153}$$

Explicitly, independently of parameters $\vartheta_1$, $\vartheta_2$, the following holds,

$$= = \top\;\top. \tag{154}$$

Applying this relation to Eq. (151), immediately reduces it to the overlap between the top and bottom vectors, which can be explicitly evaluated as

$$\langle L_{\vartheta_1} | R_{\vartheta_2} \rangle = = 1. \tag{155}$$

□

## B.2   Proof of Property 2

*Proof.* To prove Eq. (108) diagrammatically, we have to introduce an auxiliary matrix

$$S = \begin{bmatrix} 1 & & \\ & & 1 \\ & 1 & \end{bmatrix} = \text{—}\blacksquare\text{—}, \tag{156}$$

which is used to express stationarity of the MPS in terms of local relations (see [57] for the details),

$$= . \tag{157}$$

This relation allows us to prove that the MPS is stationary for a finite system with periodic boundaries. It can be also used to prove (108) when combined with the following boundary identities

$$= (1-\vartheta_{\mathrm{L}}) \qquad , \qquad = (1-\vartheta_{\mathrm{L}}) \qquad ,$$

$$= \frac{1}{1-\vartheta_{\mathrm{L}}} \qquad , \qquad = \frac{1}{1-\vartheta_{\mathrm{L}}} \qquad . \tag{158}$$

The case of $\mathcal{C}_0$ has to be treated separately, and one can directly check that the following holds,

$$
\text{(159)}
$$

The proof of Eq. (109) is analogous and follows directly from the following set of local relations

$$
\text{(160)}
$$

The first is a diagrammatic representation of the unitarity of the local time-evolution operator, while the rest are the consistency relations that have to be satisfied by the tensors of the left and the right fixed points. $\qquad\square$

### B.3 Additional details on the proof of Property 3

The $9 \times 9$ matrix $\mathcal{C}_0$ that governs the relaxation to the steady state takes the following form

$$
\mathcal{C}_0 =
\begin{bmatrix}
-\bar{\vartheta}_L\bar{\vartheta}_R & 0 & 0 & \bar{\vartheta}_L\bar{\vartheta}_R & \bar{\vartheta}_R & -\bar{\vartheta}_R & -\bar{\vartheta}_L\bar{\vartheta}_R & 0 & 0 \\
\bar{\vartheta}_L\vartheta_R & 0 & 0 & \bar{\vartheta}_L\vartheta_R & \vartheta_R & \bar{\vartheta}_R & -\bar{\vartheta}_L\vartheta_R & 0 & 0 \\
\bar{\vartheta}_L\vartheta_R & 0 & 0 & \bar{\vartheta}_L\vartheta_R & -\dfrac{\vartheta_R^2}{\bar{\vartheta}_R} & -\vartheta_R & -\bar{\vartheta}_L\vartheta_R & 0 & 0 \\
0 & 1 & 0 & 0 & 0 & 0 & 0 & 0 & 0 \\
\vartheta_L & 0 & 0 & \vartheta_L & 0 & 0 & \bar{\vartheta}_L & 0 & 0 \\
0 & 0 & 0 & 0 & 0 & 0 & 0 & 0 & 0 \\
0 & 0 & 0 & 0 & 0 & 0 & 0 & 0 & 0 \\
\vartheta_L & 0 & 0 & -\dfrac{\vartheta_L^2}{\bar{\vartheta}_L} & 0 & 0 & -\vartheta_L & 0 & 0 \\
0 & 0 & 0 & 0 & 0 & 0 & 0 & 0 & 0
\end{bmatrix}, \qquad \bar{\vartheta}_{L/R} = 1 - \vartheta_{L/R}, \quad \text{(161)}
$$

and by explicit diagonalisation we confirm that there are exactly three non-zero eigenvalues with multiplicities 1,

$$
\mathrm{Sp}(\mathcal{C}_0) = \{0, 1, \Lambda_1, \Lambda_2\}, \qquad \Lambda_{1,2} = -\frac{\vartheta_L + \vartheta_R - \vartheta_L\vartheta_R}{2}\left(1 \pm \sqrt{1 - \frac{4\vartheta_L\vartheta_R}{\vartheta_L + \vartheta_R - \vartheta_L\vartheta_R}}\right), \quad \text{(162)}
$$

with corresponding (right) eigenvectors $|1_0\rangle$ (given by (107)) and $\left|\Lambda_{1,0}\right\rangle$, $\left|\Lambda_{2,0}\right\rangle$,

$$
\left|\Lambda_{1/2,0}\right\rangle = \frac{1}{\vartheta_L(1-\vartheta_L)(1+\vartheta_R+\vartheta_L)}
\begin{bmatrix}
(1-\vartheta_L)(1-\vartheta_R)(\vartheta_L\vartheta_R + \Lambda_{1/2}) \\
\vartheta_R(1-\vartheta_L)(1-\vartheta_R)(\vartheta_L\vartheta_R + \Lambda_{1/2}) \\
(1-\vartheta_L-\vartheta_R+2\vartheta_L\vartheta_R)(\vartheta_L + \Lambda_{1/2}) \\
\vartheta_R(1-\vartheta_L)((1-\vartheta_L)(1-\vartheta_R) - \Lambda_{1/2}) \\
-(1-\vartheta_L)\big(\vartheta_L\vartheta_R^2 + (1-\vartheta_R)(\vartheta_R + \Lambda_{1/2})\big) \\
0 \\
0 \\
\vartheta_L(1-\vartheta_L-\vartheta_R+2\vartheta_L\vartheta_R) \\
0
\end{bmatrix}. \quad \text{(163)}
$$

Expanding $|\Phi_0(\Psi)\rangle$ in eigenstates of $\mathcal{C}_0$ we have

$$
|\Phi_0(\Psi)\rangle = |1_0\rangle + \gamma_1\left|\Lambda_{1,0}\right\rangle + \gamma_2\left|\Lambda_{2,0}\right\rangle, \quad \text{(164)}
$$

where the constants $\gamma_{1/2}$ are given by

$$\gamma_{1/2} = \frac{2\vartheta_L\vartheta_R - (1-\vartheta_R-\vartheta_L)\Lambda_{1/2}}{(1-\vartheta_L-\vartheta_R+2\vartheta_R\vartheta_L)(\Lambda_{1/2}-\Lambda_{2/1})}. \tag{165}$$

As explained in the main-text proof of Property 3 (see the discussion between Eqs. (117) and (118)), this allows us to express $\mathcal{C}_l^t \left|\Phi_{x,l-x}(\Psi_0)\right\rangle$ for *any* $t$ satisfying the condition

$$2t > 3m, \qquad m = \max\{x, l-x\}, \tag{166}$$

as

$$\mathcal{C}_l^t \left|\Phi_{x,l-x}(\Psi_0)\right\rangle = \left|1_l\right\rangle + \gamma_1 \Lambda_1^{t-\frac{3}{2}m} \mathcal{A}_{x,l-x} \left|\Lambda_{1,0}\right\rangle + \gamma_2 \Lambda_2^{t-\frac{3}{2}m} \mathcal{A}_{x,l-x} \left|\Lambda_{2,0}\right\rangle. \tag{167}$$

From here it follows that $\left|\Lambda_{1/2,(x,l-x)}\right\rangle$, defined as

$$\left|\Lambda_{1/2,(x,l-x)}\right\rangle = \Lambda^{-\frac{3}{2}m} \mathcal{A}_{x,l-x} \left|\Lambda_{1/2,0}\right\rangle, \tag{168}$$

are eigenvectors of $\mathcal{C}_l$, corresponding to the subleading eigenvalues $\Lambda_1$ and $\Lambda_2$, and Eq. (167) can be decomposed as

$$\mathcal{C}_l^t \left|\Phi_{x,l-x}(\Psi_0)\right\rangle = \left|1_l\right\rangle + \gamma_1 \Lambda_1^t \left|\Lambda_{1,(x,l-x)}\right\rangle + \gamma_2 \Lambda_2^t \left|\Lambda_{2,(x,l-x)}\right\rangle. \tag{169}$$

To bound the norm of eigenvectors $\left|\Lambda_{1/2,(x,l-x)}\right\rangle$, we first observe that they can be equivalently expressed as,

$$\begin{aligned}
\left|\Lambda_{1/2,(x,l-x)}\right\rangle &= \frac{\mathcal{C}_l - \Lambda_{2/1}}{\gamma_{1/2}\Lambda_{1/2}^{\frac{3}{2}m}(\Lambda_{1/2}-\Lambda_{2/1})}\left(\mathcal{C}_l^{\frac{3}{2}m}\left|\Phi_{x,l-x}(\Psi_0)\right\rangle - \left|1_l\right\rangle\right) \\
&= \frac{(\mathcal{C}_l-\Lambda_{2/1})(\mathcal{C}_l-1)}{\gamma_{1/2}\Lambda_{1/2}^{\frac{3}{2}m}(\Lambda_{1/2}-\Lambda_{2/1})(\Lambda_{1/2}-1)}\mathcal{C}_l^{\frac{3}{2}m}\left|\Phi_{x,l-x}(\Psi_0)\right\rangle,
\end{aligned} \tag{170}$$

and the prefactors on the r.h.s. of (170) are well defined also when $\Lambda_1 = \Lambda_2$. Then the norm of the eigenvector can be bounded by noting that the action of $\mathcal{C}_l$ on $\left|\Phi_{x,l-x}(\Psi_0)\right\rangle$ can be equivalently reproduced by

$$\bar{\mathcal{C}}_l = \left|\bar{1}_l\right\rangle\!\left\langle 1_l\right| + \Lambda_1 \left|\bar{\Lambda}_{1,(x,l-x)}\right\rangle\!\left\langle\Lambda_{1,(x,l-x)}\right| + \Lambda_2 \left|\bar{\Lambda}_{2,(x,l-x)}\right\rangle\!\left\langle\Lambda_{2,(x,l-x)}\right|, \tag{171}$$

with spectrum $\{1, \Lambda_1, \Lambda_2, 0\}$ (cf. (169)),

$$\begin{aligned}
\left\|\left|\Lambda_{1/2,(x,l-x)}\right\rangle\right\| &= \frac{\left\|\mathcal{C}_l^{\frac{3}{2}m}(\mathcal{C}_l-\Lambda_{2/1})(\mathcal{C}_l-1)\left|\Phi_{x,l-x}(\Psi_0)\right\rangle\right\|}{\left|\gamma_{1/2}\Lambda_{1/2}^{\frac{3}{2}m}(\Lambda_1-\Lambda_2)(\Lambda_{1/2}-1)\right|} \\
&= \frac{\left\|\bar{\mathcal{C}}_l^{\frac{3}{2}m}(\bar{\mathcal{C}}_l-\Lambda_{2/1})(\bar{\mathcal{C}}_l-1)\left|\Phi_{x,l-x}(\Psi_0)\right\rangle\right\|}{\left|\gamma_{1/2}\Lambda_{1/2}^{\frac{3}{2}m}(\Lambda_1-\Lambda_2)(\Lambda_{1/2}-1)\right|} \\
&\leq \frac{\left\|\left|\Phi_{x,l-x}(\Psi_0)\right\rangle\right\|}{\left|\gamma_{1/2}(\Lambda_1-\Lambda_2)(\Lambda_{1/2}-1)\right|},
\end{aligned} \tag{172}$$

where in the last step we used that $\bar{\mathcal{C}}_l^k(\bar{\mathcal{C}}_l-\Lambda_{2/1})(\bar{\mathcal{C}}_l-1)$ has operator norm bounded by $\Lambda_{1/2}^k$ for each $k \geq 1$. As there is no $l$-dependence in the bound on the eigenvectors (apart from the norm of $\left|\Phi_{x,l-x}(\Psi_0)\right\rangle$), the scaling regime discussed in Subsection 5.2.2 is well defined.

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
