# Peer review of "Exact relaxation to Gibbs and non-equilibrium steady states in the quantum cellular automaton Rule 54"

_SciPost Physics, doi:SciPost Phys. 11, 106 (2021)_

## Round 1 · Referee Report · Anonymous (Referee 1) · 2021-7-28

Strengths

1- Rigorous arguments 2- Use of diagrammatic representation 3- Novelty of results

Report

The authors consider a specific case of out of equilibrium dynamics, modeled by the QCA rule 54, to identify a class of states for which a non equilibrium steady state is reached and an exact characterization can be provided. The analysis is perfored by adopting a general time-channel description for the dynamics of local operators. The authors introduce and intensively exploit a diagrammatic state representation to demostrate the one-parameter family of Gibbs states reached after relaxation.
The analysis is rigorously performed and all the statements are extensively explained.
I only point out a typo on page 19 (pugging ->plugging)
In my opinion this work fully meets the general acceptance criteria. Moreover, it presents a breakthrough on a previously-identified and long-standing research stumbling block, namely the characterization of dynamics originating non equilibrium steady states. It also could bridge between studies about thermalization in out of equilibrium systems and matrix product states numerical simulations. For these reasons, I suggest publication in the current form of the draft.

---

## Round 1 · Referee Report · Anonymous (Referee 2) · 2021-9-16

Strengths

1) Formally correct. 2) In some cases it makes a direct connection between exact calculations of the dynamics and the generalized hydrodynamics in an interacting model.

Weaknesses

1) Not easy to read for non specialists.
2) General discussion of Rule 54 model is missing.

Report

In the work 'Exact relaxation to Gibbs and non-equilibrium steady states in the quantum cellular automaton Rule 54', K. Klobas and B. Bertini analyze the the out-of-equilibrium dynamics of the quantum cellular automaton Rule 54.

The paper presents in a detailed form the technical calculations, mostly of diagrammatic nature, some aspects of the dynamics of the so-called 'Rule 54' model. The authors consider a class of initial states for which they are able to provide an explicit construction of the fixed-points of the space transfer matrix. This is used as a tool to investigate their time-evolution and relaxation in terms of finite-dimensional quantum maps. For the class of initial states considered, relaxation takes place (somewhat unexpectedly) exponentially fast to Gibbs states. For most of the cases, these results agree with the predictions of generalized hydrodynamics, providing a confirmation of the GHD for an inhomogeneous quench in an interacting system.

The paper is technical and it requires a detailed reading to understand the diagrammatic approach. I have three main comments: - This paper is the first of a series of two works by the same authors. For this reason, I would strongly recommend to add a couple of additional paragraphs at the beginning of sec.3, or alternatively in the introduction, to discuss Rule 54 model with a broader perspective (not only citing relevant references) which would be of great importance for non experts in this technical field. I am also convinced that this would improve much the readability of the paper(s). - In Sec.5 the authors derive relevant relations for inhomogeneous quenches. The author might consider to move the proof of property 3 to the appendices and leave the relevant physical discussion about the exponential relaxation in the main text. - A minor point: ref. [79] is incorrect. It cites Bobenko et al. but it links to a different PRL.

I would also recommend the authors to check the English as I found a few misspelled words.

In summary, the results are sound, the proofs are detailed and the findings are clear. After a consideration of the suggestions above, I would recommend this work for publication.

Requested changes

1) This paper is the first of a series of two works by the same authors. For this reason, I would strongly recommend to add a couple of additional paragraphs at the beginning of sec.3, or alternatively in the introduction, to discuss Rule 54 model with a broader perspective (not only citing relevant references) which would be of great importance for non experts in this technical field. I am also convinced that this would improve much the readability of the paper(s). 2) In Sec.5 the authors derive relevant relations for inhomogeneous quenches. The author might consider to move the proof of property 3 to the appendices and leave the relevant physical discussion about the exponential relaxation in the main text. 3) A minor point: ref. [79] is incorrect. It cites Bobenko et al. but it links to a different PRL.

---

## Round 2 · Referee Report · Anonymous · 2021-11-7

Report

As from my previous report, the work can be published.

---

## Round 2 · Referee Report · Anonymous · 2021-11-7

Report

I am satisfied with the modifications of the manuscript and with the detailed reply to my comments. The authors provided convincing explanations of the points raised in my first report. I therefore recommend the paper for publication in SciPost.

---

## Round 2 · Author Response

We thank both referees for their careful reading of our manuscript, for their relevant comments, and for their positive assessment. We have made a number of modifications to the manuscript to accommodate the comments of Referee 2 and to follow an interesting comment of Referee 1 of the companion paper. To help identifying the changes we highlighted them in red in the new version.

Response to referee 1

We sincerely thank the referee for their very positive assessment. We corrected the typo that they pointed out.

Response to referee 2

We thank the referee for their time and overall positive assessment. However, we believe that they did not entirely appreciate the novelty and importance of the paper's results and we find some of their marks unreasonably low. We provide a detailed response below.

We begin by commenting on some of the referee's judgements concerning strengths and weaknesses.

S2 In some cases it makes a direct connection between exact calculations of the dynamics and the generalized hydrodynamics in an interacting model.

We believe that this judgment (as well as the analogous one presented in the summary) is not entirely giving justice to the results of the paper. Our work presents the first known instance where some predictions of GHD (as well as some statements concerning relaxation after homogeneous quenches) could be rigorously proven in the presence of interactions. This is of course done in some cases because, for such a microscopic proof, one needs to specify a particular rule for the microscopic dynamics and some particular initial states. Moreover, as discussed in Sec. 6, some regions of the profiles could not be accessed. Nevertheless, we think that this is a very remarkable achievement, which is not entirely conveyed by the above statement.

W1 Not easy to read for non specialists.

As mentioned in the response to S2 this paper aims at providing (unprecedented) microscopic derivations for some conjectures concerning relaxation in interacting quantum many-body systems. This requires some level of technical detail. To make the results of the paper as accessible as possible we decided to use the well known diagrammatic language of tensor network theory, see, e.g., Ref. [62].

Even though the referee probably does not share this opinion, we believe that presenting our results using the aforementioned diagrammatic approach makes the paper easier to read for a sizeable community of theoreticians (this is, for instance, the case of Referee 1). Therefore, we decided to stick to this choice.

To further improve readability we modified the discussion in Sec. 2.

W2 General discussion of Rule 54 model is missing.

In this paper the RCA Rule 54 is used merely as a particularly simple interacting quantum circuit. To avoid overloading the reader with inessential information, we didn't include a detailed description of the many interesting properties of the model. However, we partially took this point of the referee and in the new version we expanded the discussion about Rule 54 (within the reasonable space limits for a paper that is already long (further detail can be found, e.g., in the recent review [57]).

Next, let us briefly answer the referee's main comments.

1 This paper is the first of a series of two works by the same authors. For this reason, I would strongly recommend to add a couple of additional paragraphs at the beginning of Sec.3, or alternatively in the introduction, to discuss Rule 54 model with a broader perspective (not only citing relevant references) which would be of great importance for non experts in this technical field. I am also convinced that this would improve much the
readability of the paper(s).

As discussed in the response of W2, we decided to partially follow the referee’s suggestion concerning this point.

2 In Sec.5 the authors derive relevant relations for inhomogeneous quenches. The author might consider to move the proof of property 3 to the appendices and leave the relevant physical discussion about the exponential relaxation in the main text.

After careful consideration we decided not to follow this suggestion. The proof of Property 3 is one of the main results of the paper and gives important physical insights. Therefore, we believe that it belongs in the main text. Moreover, some of the concepts introduced there are also used in the following subsections.

3 A minor point: ref. [79] is incorrect. It cites Bobenko et al. but it links to a different PRL.

Fixed. We thank the referee for pointing this out.

4 I would also recommend the authors to check the English as I found a few misspelled words.

After a very careful double-check we found 4 typos. They are all corrected in the new version.

Finally, let us briefly comment on the marks given by the referee as we find them unjustifiably low.

Validity: High

This paper is about rigorous results. Unless the referee can point out anything wrong in the derivation we would expect to see the maximal mark in this entry.

Significance: Good

We struggle to understand this mark. This paper provides the first ever proof of relaxation in an inhomogeneous setting. In the words of Referee 1 "it presents a breakthrough on a previously-identified and long-standing research stumbling block".

Originality: Good

We disagree also with this mark as we believe that the proposed discussion of the time-channel approach is, in fact, rather transparent. As discussed in the response to W1, this low mark is probably due to the referee’s lack of appreciation for the diagrammatic representation employed in the paper.

Grammar: Good

Can the referee point out an example of bad English grammar for giving such a low score? Note that "grammar" is not the same as typos.

---

## Round 2 · List of Changes

- Improved the presentation in Sec. 2.
- Added a paragraph about time-channel approaches in integrable systems in Sec. 2.
- Added a short discussion of the model at the beginning of Sec. 3.
- Extended Conclusions.
- Minor changes across the paper (typos etc).

All changes are for clarity denoted by red in the new version.

---

## Editorial Decision

published